# ARRDC5 expression is conserved in mammalian testes and required for normal sperm morphogenesis

Mariana I. Giassetti[1,2], Deqiang Miao[1,2], Nathan C. Law [1,3], Melissa J. Oatley[1,2], Julie Park [1,2], LeeLa D. Robinson[1,2], Lisette A. Maddison [1], Miranda L. Bernhardt[1] & Jon M. Oatley [1,2] ✉

In sexual reproduction, sperm contribute half the genomic material required for creation of offspring yet core molecular mechanisms essential for their formation are undefined. Here, the α-arrestin molecule arrestin-domain containing 5 (ARRDC5) is identified as an essential regulator of mammalian spermatogenesis. Multispecies testicular tissue transcriptome profiling indicates that expression of *Arrdc5* is testis enriched, if not specific, in mice, pigs, cattle, and humans. Knockout of *Arrdc5* in mice leads to male specific sterility due to production of low numbers of sperm that are immotile and malformed. Spermiogenesis, the final phase of spermatogenesis when round spermatids transform to spermatozoa, is defective in testes of *Arrdc5* deficient mice. Also, epididymal sperm in *Arrdc5* knockouts are unable to capacitate and fertilize oocytes. These findings establish ARRDC5 as an essential regulator of mammalian spermatogenesis. Considering the role of arrestin molecules as modulators of cellular signaling and ubiquitination, ARRDC5 is a potential male contraceptive target.

Sperm are the vessel for the transmission of male genetic contributions across generations and are the only mammalian cell type that performs its function outside of the body in which it was produced[1,2]. For natural fertility, millions of sperm that possess properly organized haploid genomes must be generated per day through the process of spermatogenesis[3]. When formed normally, sperm can navigate the female reproductive tract via motility from a specialized flagellum, undergo capacitation as a final step in attaining the capacity to bind an ovulated oocyte, and finally fertilize the oocyte by fusion of the head with the oocyte plasma membrane that initiates delivery of the haploid paternal genome. Deficiencies in the ability to generate sufficient numbers of sperm (referred to as oligospermia), motile sperm (referred to as asthenospermia), or morphologically normal sperm (referred to as teratospermia) manifest as male subfertility or infertility[4–6]. When a male possesses all these deficiencies, the condition is referred to as oligoasthenoteratospermia or OAT. Incidences of this condition leading to fertility defects can be found throughout mammalian species, including domestic livestock[7,8], endangered wildlife[9], and humans[10]. At present, the core molecular mechanisms that are evolutionarily conserved across Mammalia for the regulation of normal spermatogenesis and molecular deficiencies that could cause OAT are undefined.

Spermatogenesis is the process by which haploid sperm are formed from diploid spermatogonia and involves three major phases; amplification of spermatogonia via mitotic divisions, two meiotic divisions of spermatocytes that produce haploid round spermatids, and spermiogenesis by which round spermatids transform into specialized elongated spermatids[3,11]. During spermiogenesis, post-meiotic round spermatids undergo a process of elongation, chromatin compaction, repositioning of their mitochondria, and morphologically transforming to create a specialized cell type with a head, neck, midpiece, and tail[12]. A histone-to-protamine exchange process leads to a highly compacted haploid genome that influences head shape and

[1]Center for Reproductive Biology, Washington State University, Pullman, WA, USA. [2]School of Molecular Biosciences, College of Veterinary Medicine, Washington State University, Pullman, WA, USA. [3]Department of Animal Sciences, Washington State University, Pullman, WA, USA. ✉e-mail: joatley@wsu.edu

biogenesis of an acrosome containing proteolytic enzymes needed for fertilization[13–15]. The neck structure aligns the head with the midpiece and tail to create a streamlined cell. The midpiece and tail are formed by an axoneme with 9 + 2 microtubule cytoskeleton to create a powerful flagellum that is needed for motility[16]. The energy required for beating the sperm flagellum is provided by mitochondria that are helically wrapped around the outer dense fibers of the midpiece and glycolytic activity localized to the fibrous sheath[16]. Each of these components must be assembled correctly during the process of spermiogenesis to yield a sperm cell that can navigate the female reproductive tract to find and fertilize an oocyte. Defects manifest as OAT, which is a predominant diagnosis of male fertility disorders[10].

The arrestin family of proteins is an ancient clan with a diverse array of functions in yeast to humans[17]. The first described molecules in this family were termed visual and β-arrestins that possess C-terminal tails containing binding sites for clathrin and AP2[18]. Over the last decade, a second closely related group termed α-arrestins, or arrestin-domain-containing (ARRDC) proteins, has been described in a variety of eukaryotes which differ from the visual/β-arrestins in the N- and C-terminal domains[19–21]. In mammals, ten arrestin proteins have been described, including four visual/β-arrestins and six α-arrestins. Initially discovered as desensitizers of G protein-coupled receptor signaling, the array of functions for arrestin family proteins has been expanded to include serving as adapters for E3 ubiquitin ligases to facilitate the penultimate step in protein ubiquitination[20,22]. At present, understanding the role that arrestin family proteins may have in regulating spermatogenesis is rudimentary. A previous study reported enriched expression of ARRDC4 in the epididymis of mice and genetic inactivation resulted in reduced sperm motility and a subfertility phenotype[23]. Beyond this single study, the expression of other arrestin proteins in mammalian testes and functional assessment of possible importance in regulating spermatogenesis have not been described.

In the present study, we generated an integrated single-cell transcriptome profile of testicular tissue from mice, cattle, and pigs to identify gene expression in germ cells that is potentially conserved across mammalian species. Through bioinformatic analysis, a refined list of candidate genes with germ cell-enriched expression that had not been reported previously in the peer-reviewed scientific literature to be linked functionally with spermatogenesis was generated. From this list, the α-arrestin protein-encoding gene *Arrdc5* emerged as a prime candidate. In mice, pigs, and cattle, *Arrdc5* mRNA is detectable in testes only, and expression is enriched in the testes of humans. To assess the biological function of *Arrdc5*, CRISPR-Cas9 gene editing was used to generate mice with null alleles. Through phenotyping analysis, the impact of ARRDC5 deficiency was found to be male-specific sterility due to significant reductions in sperm numbers and motility and increased abnormal sperm morphology. Other phenotypic abnormalities of *Arrdc5* null mice were not observed and knockout females were found to have normal fertility. Collectively, these results reveal the importance of arrestin proteins in germ cell function and demonstrate a specific role for ARRDC5 in regulating spermatogenesis with genetic deficiency leading to OAT and male infertility.

## Results

### Integrated multispecies testicular single-cell transcriptome database

To identify gene expression in germ cells that is conserved across mammalian species, we used single-cell RNA-sequencing (scRNA-seq) with the 10X-Genomics and Illumina platforms to analyze transcriptome profiles of testicular tissue from mice, cattle, and pigs (Fig. 1a). Because the abundance of haploid spermatids in adult testes can overshadow the depth of analysis for transcriptionally active diploid germ cells, tissue from males at the prepubertal phase of development were used. Following quality control filtering, transcriptomes for ~23,000 bovine testis cells, ~22,000 porcine testis cells,

and ~3000 murine testis cells were generated for analysis (Fig. S1). Through Uniform Manifold Approximation and Projection (UMAP) analysis, the cells from all three species were then computationally combined as an integrated dataset (Figs. 1b and S2a). This analysis revealed distinct clusters that could be assigned specific testicular cell type identities (Germ Cell, Sertoli Cell, Leydig Cell, Myoid Cell, Lymphocyte, and Macrophage) based on the expression of canonical biomarker genes (Figs. 1b and S2b–g). In total, 3658 germ cells, 16,997 Sertoli cells, 5988 Leydig cells, 15,533 myoid cells, 5057 lymphocytes, and 2245 macrophages could be identified. This integrated multi-species single-cell transcriptome database for several key testicular cell types provides a unique and potentially invaluable resource for mining to identify both evolutionarily conserved and species-specific gene expression.

### Identification of conserved gene expression in testicular germ cells

Next, we used the integrated single-cell transcriptome database to identify gene expression in germ cells that is conserved across the three mammalian species profiled. Based on the Venn diagram analysis of the germ cell cluster defined by UMAP analysis, 10,183 genes were found to be co-expressed by cattle, pigs, and mice (Fig. 1c and Supplementary Data 1). Gene ontology analysis revealed that these conserved germ cell-expressed genes encode for molecules with a variety of functional attributes, with the greatest enrichment being the ubiquitin-proteasome pathway and transcriptional regulation (Fig. S3). Next, we developed a bioinformatics filtering pipeline to identify evolutionarily conserved candidate molecules that could be essential regulators of spermatogenesis (Fig. 1d). First, the list of 10,183 conserved germ cell-expressed genes was cross-referenced with the international mouse phenotyping consortium database to select for those that have an inferred function in male mouse fertility which yielded 106 genes (Supplementary Data 1). Second, we filtered the list of 106 genes for those with testis-enriched expression based on cross-referencing with the ENCODE and human Genotype-Tissue Expression (GTEx) portal, which yielded 14 genes (Supplementary Data 1). Third, this shortened list was then cross-referenced back to the integrated multispecies testicular single-cell transcriptome database to define those with presumptive germ cell-specific expression, which yielded nine genes (Supplementary Data 1). Of these, a biological role for the gene arrestin-domain-containing 5 (*Arrdc5*) had not been described for any cell type in the peer-reviewed scientific literature.

### Testis-specific expression of *Arrdc5* in mammalian spermatogenesis

The bioinformatics analysis of our multispecies integrated scRNA-seq dataset suggested that expression of the *Arrdc5* gene is enriched in testicular tissue with presumptive germ cell specificity. Outcomes of RT-PCR analysis with several different tissues from cattle, pigs, and mice revealed that *Arrdc5* mRNA is detectable in testes only (Fig. 2a–c), which confirmed data from the ENCODE project tissue-level gene expression in mice (Fig. S4a). Notably, *Arrdc5* transcript was detectable by RT-PCR analysis in testes from adult wild-type mice but not *Nanos2* knockout mice that lack germ cells (Fig. 2a). Similarly, data from the human tissue GTEx portal indicates that *Arrdc5* expression is highly enriched in testes and low to undetectable in other tissues and cell types (Fig. S4B). Together, these findings suggest that the expression of *Arrdc5* in mammalian species from mice to humans is germ cell-specific.

The germ cell populations of prepubertal mice, cattle, and pigs used for the generation of the multispecies integrated scRNA-seq dataset included spermatogonia as well as early prophase spermatocytes for the pig tissue (Figs. 1 and S2). Further assessment of the germ cell cluster revealed that *Arrdc5* transcript could be detected in a small percentage of the population (Fig. S5a, b). To gain further insight into the temporal expression of *Arrdc5* throughout spermatogenesis, we

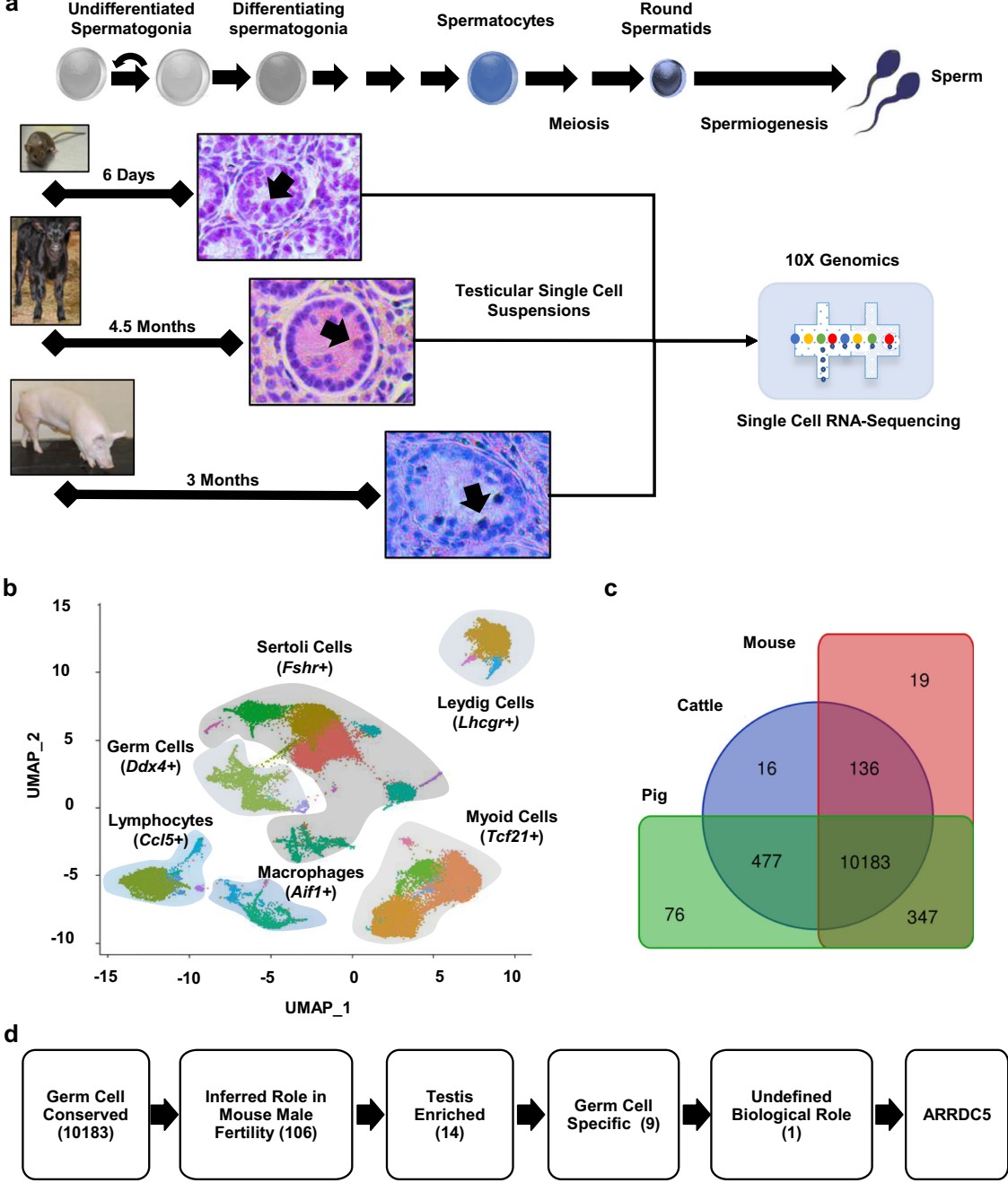

**Fig. 1 | Generation of a multispecies integrated testicular single-cell transcriptome database. a** Schematic of the experimental strategy that analyzed testicular tissue from prepubertal mice, cattle, and pigs by single-cell RNA-sequencing. Representative images of hematoxylin and eosin-stained testicular cross-sections are shown, arrows denote germ cells within seminiferous tubules. **b** Integrated uniform manifold approximation and projection (UMAP) plot for ~48,000 testicular cells from three species. Distinct clusters of testicular cell types were defined based on the expression of canonical biomarker genes. **c** Venn diagram analysis of gene expression in the germ cell cluster based on species identity, which yielded 10,183 genes as having conserved expression across mouse, cattle, and pig. **d** Bioinformatic filtering pipeline to identify novel evolutionarily conserved candidate molecules that could be essential regulators of spermatogenesis. The number of genes selected at each step are in parentheses.

mined previously published scRNA-seq data of testicular tissue from mice at postnatal ages ranging from birth to adulthood[24–26]. While *Arrdc5* transcripts were detectable in spermatogonia at all ages, major upregulation occurred at 20–30 days of age when secondary spermatocytes, as well as haploid round spermatids, first arise in the germline (Fig. S5c, d). Notably, *Arrdc5* mRNA upregulation was found to align partially with that of *Tnp1* and *Prm1* mRNA, genes that are expressed in spermatocytes and stored for translation in round

spermatids (Fig. S5e, f). However, in contrast to *Tnp2* and *Prm1* expression, which persists at P30 and P56 when condensing and elongated spermatids arise, *Arrdc5* expression is downregulated at these phases of spermatogenesis (Fig. S5d). Together, these findings suggest that nascent transcription initiates in spermatogonia and persists through meiotic spermatocytes.

Next, we aimed to define the testicular cell types in which the ARRDC5 protein is present. First, immunostaining of testicular

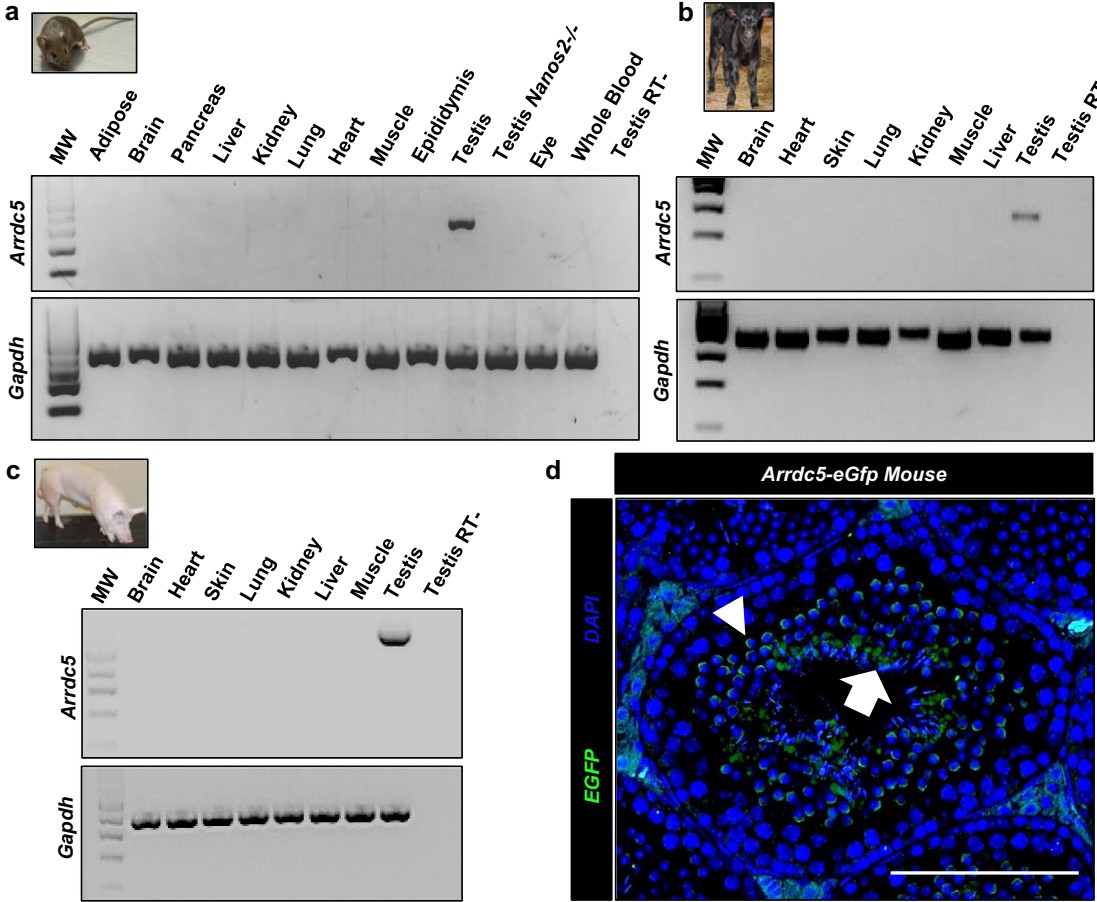

**Fig. 2 | Arrdc5 expression profile in mammalian species. a** RT-PCR analysis of *Arrdc5* mRNA in mouse tissues. MW is 100 bp DNA ladder and *Gapdh* was used as a loading control. *Nanos2−/−* testes are ablated of germ cells. **b** RT-PCR analysis of *Arrdc5* mRNA in cattle tissues. MW is a 100 bp DNA ladder and *Gapdh* was used as a loading control. **c** RT-PCR analysis of *Arrdc5* mRNA in pig tissues. MW is a 100 bp DNA ladder and *Gapdh* was used as a loading control. **d** Seminiferous tubule cross-sections from testes of adult *Arrdc5-eGfp* mice that were immunostained for EGFP (green color) to assess cell type expression of Arrdc5 protein. DAPI (blue color) was used to stain DNA. Arrow indicates elongated spermatids and the arrowhead indicates stained round spermatids. Bar is 50 μm. All images are representative of three independently repeated experiments.

cross-sections from several species was attempted with commercially available antibodies but did not yield reliable detection (Fig. S6). To overcome this obstacle, a mouse model was generated using CRISPR-Cas9 technology to knock-in enhanced green fluorescent protein (eGFP) coding sequence into the *Arrdc5* locus (Fig. S7a, b). The strategy replaced the stop codon of exon 3 and included a P2A peptide sequence that liberates EGFP from ARRDC5 during translation. As such, imaging of EGFP provides an accurate assessment of the ARRDC5 protein expression profile. Germline transmission of the *Arrdc5-eGfp* allele occurred at expected Mendelian ratios and both heterozygous and homozygous animals were viable and fertile. In testes of adult *Arrdc5-eGfp* mice, EGFP was detectable in germ cells at the round and elongated spermatid phases of development (Figs. 2d and S7c, S8). Collectively, these findings suggest that although *Arrdc5* gene transcription occurs in spermatogonia and spermatocytes (based on scRNA-seq profiles), like the expression of transition proteins (*Tnp1*, *Tnp2*) and protamines (*Prm1* and *Prm2*), the protein is not translated until the haploid round spermatid phase of spermatogenesis.

Arrestin molecules are an ancient family found throughout Eukarya, from yeast to humans. In Metazoans, members include visual/β-arrestins, VPS26-like proteins, and α-arrestins (i.e., ARRDCs). Of these, the α-arrestins are the least characterized and have not been studied in testicular tissue. Beyond primates, *ARRDC5* orthologs are present in the genomes of livestock, rodents, monotremes, birds, and reptiles (Fig. S9). Based on phylogenetic analysis, the *ARRDC5* genes in

pigs and cattle are more closely related to the primate gene than in other placental mammals. Indeed, pig and cattle *ARRDC5* protein sequences share 77 and 75% with human *ARRDC5*, respectively (Fig. S9). Together, these data suggest that *ARRDC5* is evolutionarily conserved across Animalia with testis-specific expression.

## ARRDC5 is required specifically for male fertility in mice

Considering the evolutionarily conserved nature of *ARRDC5* and testis-enriched expression in mice, pigs, cattle, and humans, we next aimed to determine whether it plays an important role in spermatogenesis. To achieve this, CRISPR-Cas9 technology was used to engineer mice with inactivated *Arrdc5* alleles by direct manipulation and transfer of C57BL/6J;129SvlmJ hybrid embryos (Fig. S10a–c). From this process, a founder male possessing an *Arrdc5* allele with a 308 bp deletion in exon 1 that removed the start codon was generated (Fig. S10c). Breeding of the *Arrdc5^{308Δ/+}* founder male to wild-type C57BL/6J females resulted in germline transmission and the N1 offspring were crossed for filial breeding to create an experimental line for analysis. Mating of *Arrdc5^{308Δ/+}* mice produced *Arrdc5^{308Δ/Δ}* at expected Mendelian ratios, which were viable from birth through adulthood (Fig. 3a). Based on RT-PCR analysis, *Arrdc5* transcript could not be detected in testes of mice homozygous for the 308 bp Δ allele (Fig. S10d), thus confirming a null state had been engineered, designated hereafter as *Arrdc5−/−*. At both 3 weeks and 16 weeks of age, the body weights of *Arrdc5−/−* and wild-type *Arrdc5+/+* male littermates were found to be

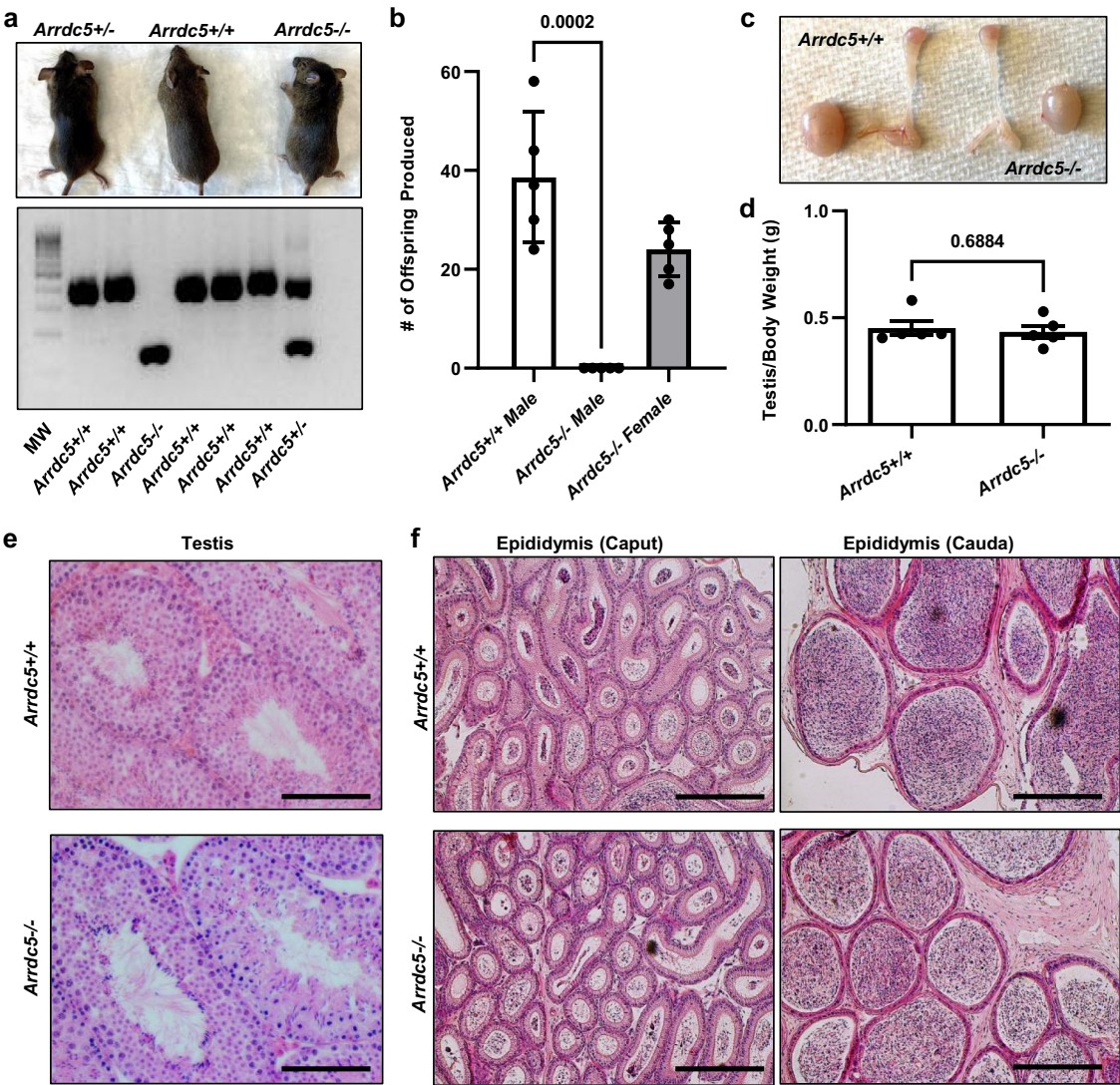

**Fig. 3 | Impact of *Arrdc5* genetic inactivation in mice. a** Male mice heterozygous (+/−), homozygous (−/−), or non-possessing (+/+) of an inactivating 308 bp deletion allele for the *Arrdc5* coding sequence, and an agarose gel for visualization of genomic DNA genotyping analysis. MW is a 100 bp DNA ladder. **b** Quantitation of fecundity for adult *Arrdc5*−/− male and female mice as well as control *Arrdc5*+/+ male littermates over a 4-month period of pairing with wild-type female or male mice. Data bars are mean ± SEM of the total numbers of pups born and dots represent values for individual animals (*n* = 5 for each genotype). **c** Testes and epididymis from adult *Arrdc5*−/− and *Arrdc5*+/+ littermates. **d** Quantitation of the testis/body weight ratio for adult *Arrdc5*−/− and *Arrdc5*+/+ littermates. Data bars are mean ± SEM and dots represent values for individual animals (*n* = 5 for each genotype). **e** Hematoxylin and eosin-stained cross-sections from testes of adult *Arrdc5*−/− and *Arrdc5*+/+ littermates. Bars are 100 μm. **f** Hematoxylin and eosin-stained cross-sections from the head (caput) and tail (cauda) of epididymis from adult *Arrdc5*−/− and *Arrdc5*+/+ littermates. Bars are 100 μm. Images are representative of >10 (**a**) and 5 (**c, e, f**) independently repeated experiments. For quantitative comparisons, differences between genotypes were analyzed statistically using unpaired two-tailed *t*-tests. Source data are provided as a Source Data file.

no different (Fig. S10e). The pairing of *Arrdc5*−/− male mice (*n* = 5) with pubertal wild-type females for a 4-month period yielded zero pups, but copulation occurred as evidence by the presence of vaginal plugs, whereas *Arrdc5*+/+ littermates (*n* = 5) had an expected level of fecundity when paired with wild-type females (Fig. 3b). In addition, *Arrdc5*−/− female mice (*n* = 5) paired with wild-type males generated a number of offspring over a 2–3-month period that was within the range of normal wild-type females (Fig. 3b). These findings demonstrate that ARRDC5 function is required specifically for male fertility and suggest that deficiency does not have an overt negative impact on other physiological systems that impact health.

**Oligoasthenoteratospermia in mice with genetic inactivation of *Arrdc5***

To investigate the cause of sterility in *Arrdc5*−/− male mice, we collected testes and epididymides from adults (2–4 months of age)

for histological analysis. Gross abnormalities of either organ were not evident (Fig. 3c), and testis size was quantitatively not different compared to *Arrdc5*+/+ littermates (Fig. 3d). In addition, the wet weight of seminal vesicles, which is correlated to serum testosterone levels, was not different between *Arrdc5*−/− and *Arrdc5*+/+ mice (Fig. S11a). Basic histological assessment of testis cross-sections revealed intact seminiferous tubules and epithelium with seemingly normal spermatogenesis, including properly organized layers of spermatogonia, spermatocytes, and round/elongating spermatids and differences compared to testes from *Arrdc5*+/+ littermates were not observable (Fig. 3e). In addition, morphological differences in cross-sections of epididymides from *Arrdc5*−/− and *Arrdc5*+/+ littermates were not evident (Fig. 3f). These observations demonstrate that postnatal establishment of the seminiferous epithelium and maintenance in adulthood is not disrupted in the absence of ARRDC5.

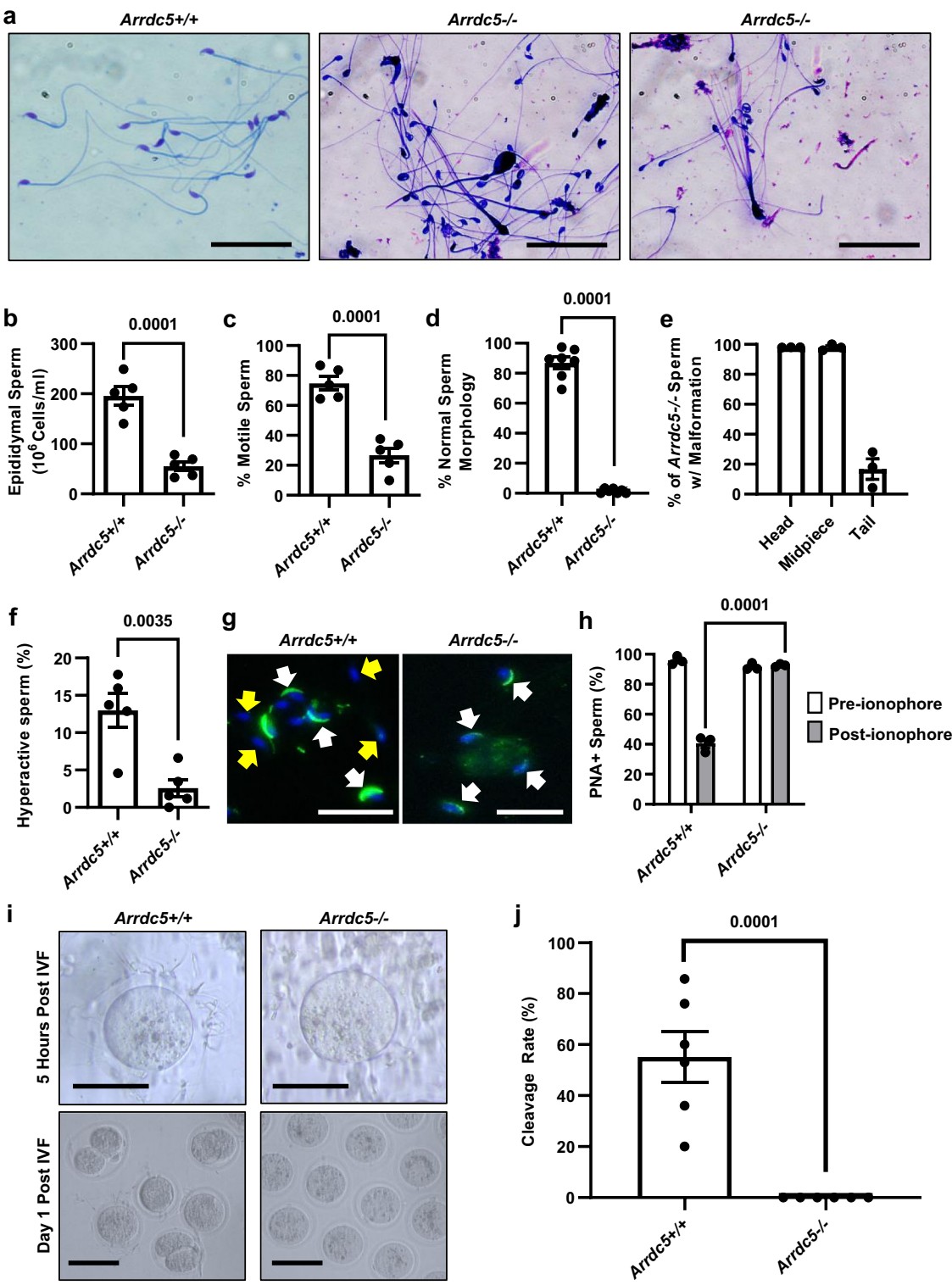

Next, we collected epididymis from adult *Arrdc5−/−* and *Arrdc5+/+* littermates for sperm parameter analysis. In contrast to the histological analysis of testes and epididymis, examination of epididymal sperm from *Arrdc5−/−* males revealed several morphological abnormalities that were not observed in *Arrdc5+/+* littermates including malformed heads, bent necks, misshapen midpieces, shortened flagella, multiple flagella, and enlarged sperm (Figs. 4a and S11b). Based on computer-assisted sperm analysis (CASA), multiple parameters were found to be significantly different for epididymal sperm from *Arrdc5−/−* males compared to *Arrdc5+/+* littermates (Fig. 4b–e). The concentration of epididymal sperm in *Arrdc5−/−* males ($n = 5$ different mice) was significantly ($P = 0.0001$) reduced to 28% of normal (Fig. 4b). In addition, the percentage of motile sperm for *Arrdc5−/−* males ($n = 4$–5 different mice) was calculated to be $26.7 \pm 4.9\%$ (mean ± SEM) which was significantly ($P = 0.0001$) different by 2.8-fold compared to *Arrdc5+/+* littermates ($n = 4$ different mice) in which $74.8 \pm 4.5\%$ (mean ± SEM) of sperm were motile (Fig. 4c). Also, the percentage of morphologically normal sperm from *Arrdc5−/−* males ($n = 7$ different mice) was calculated to be only $1.4 \pm 0.4\%$ (mean ± SEM) which was significantly ($P < 0.0001$) different by 62.1-fold compared to *Arrdc5+/+* littermates

**Fig. 4 | Sperm parameter assessment of *Arrdc5*−/− mice. a** Dip Quick stained epididymal sperm from adult *Arrdc5*+/+ and *Arrdc5*−/− littermates. Bars are 50 μm. **b**–**d** Quantitative comparison of epididymal sperm concentration **b**, motility **c**, normal morphology **d** between adult *Arrdc5*+/+ and *Arrdc5*−/− littermates. Data bars are mean ± SEM and dots represent values for individual animals (**b**, **c**: *n* = 5 for each genotype; **d**: *n* = 7 for each genotype). **e** Categorization of morphological abnormalities for epididymal sperm from adult *Arrdc5*−/− mice. Data bars are mean ± SEM and dots represent values for individual animals (*n* = 3). **f** Quantitative comparison of epididymal sperm hyperactivation between adult *Arrdc5*+/+ and *Arrdc5*−/− littermates following in vitro capacitation induction. Data bars are mean ± SEM and dots represent values for individual animals (*n* = 5 for each genotype). **g** Epididymal sperm heads from adult *Arrdc5*+/+ and *Arrdc5*−/− littermates fluorescently stained for peanut agglutinin (PNA) to assess acrosome intactness. Yellow arrows indicate acrosome-reacted sperm heads without PNA staining following in vitro capacitation induction. White arrows indicate acrosome intact sperm heads with PNA staining. Bars are 20 μm. **h** Quantitative comparison of epididymal sperm from adult *Arrdc5*+/+ and *Arrdc5*−/− littermates with detectable PNA binding before and after Ca2+ ionophore treatment to induce the acrosome reaction. Data bars are mean ± SEM and dots represent values for individual animals (*n* = 3 for each genotype). **i** Oocytes after 5 h or 1 day of in vitro fertilization (IVF) with epididymal sperm from adult *Arrdc5*+/+ or *Arrdc5*−/− littermates. Bars are 100 μm. **j** Quantitative comparison of embryo cleavage rate following in vitro fertilization of wild-type zona-pellucida intact oocytes with epididymal sperm from adult *Arrdc5*+/+ or *Arrdc5*−/− littermates. Data bars are mean ± SEM and dots represent values for individual animals (*n* = 6 for each genotype). Images are representative of 5 (**a**), 3 (**g**), and 6 (**i**) independently repeated experiments. For quantitative comparisons, differences between genotypes were analyzed statistically using unpaired two-tailed *t*-tests (**b**–**d**, **f**, **j**) or two-way ANOVA with multiple comparisons (**h**). Source data are provided as a Source Data file.

(*n* = 7 different mice) in which 86.9 ± 3.8% (mean ± SEM) of sperm were of normal morphology (Fig. 4d). Notably, quantification of morphological abnormalities revealed that ~98% of sperm from *Arrdc5*−/− males have malformed heads and midpiece whereas ~17% also possessed tail abnormalities (Fig. 4e). Furthermore, the average head size of sperm from *Arrdc5*−/− males was found to be significantly increased by 1.4-fold compared to sperm from *Arrdc5*+/+ littermates (Fig. S11c). Moreover, as expected from the array of abnormal morphologies, high-magnification imaging of sperm heads stained with hematoxylin and eosin suggested acrosome malformations (Figs. S11d and S12). Collectively, these results demonstrate that ARRDC5 is required for the generation of morphologically normal sperm and genetic deficiency leads to OAT and infertility.

## Defective capacitation and acrosome reaction with ARRDC5-deficient sperm

Considering the array of abnormalities observed with sperm from ARRDC5-deficient mice, the cause of sterility could be multifaceted, including defective capacitation or inability to fuse with the oocyte plasma membrane at fertilization. Capacitation is an essential process in maturation signified by biochemical modification of the sperm head membrane to facilitate the acrosome reaction and, in parallel, hyperactivation of the sperm tail occurs; both processes are needed to navigate the female reproductive tract and penetrate the zona pellucida. To further understand the major cause, we used epididymal sperm collected from *Arrdc5*−/− and *Arrdc5*+/+ littermates (*n* = 6 different males of each genotype) for in vitro fertilization (IVF) with oocytes from wild-type females. First, we investigated the ability of sperm from *Arrdc5*−/− males to undergo capacitation which occurs naturally during navigation of the female reproductive tract and can be initiated in vitro as a final step in maturation to achieve fertilization. Following a 1-h incubation in human-tubal fluid (HTF) containing BSA, 2.6 ± 1.2% (mean ± SEM; *n* = 3 different males) of sperm from *Arrdc5*−/− mice were hyperactivated, which was significantly different by 5.1-fold compared to the 13.0 ± 2.3% (mean ± SEM; *n* = 3 different males) of hyperactivated sperm from *Arrdc5*+/+ littermates (Fig. 4f). To assess the presence of acrosomes, we incubated epididymal sperm from *Arrdc5*−/− and *Arrdc5*+/+ littermates with fluorescently labeled peanut agglutinin (PNA) which normally binds to components of intact vesicles. Imaging by fluorescent microscopy revealed *Arrdc5*−/− sperm heads with morphologically normal acrosomes similar to wild-type sperm as well as heads with abnormal acrosome morphology but still with observable PNA binding (Fig. S12). Next, we tested for the acrosome reaction by incubating sperm in HTF media supplemented with Ca2+ ionophore, followed by incubation with fluorescently labeled PNA. Prior to ionophore treatment, 95.9 ± 1.6% (mean ± SEM, *n* = 3 different males) and 91.6 ± 1.3% (mean ± SEM, *n* = 3 different males) of sperm from *Arrdc5*+/+ and *Arrdc5*−/− males had observable PNA binding, respectively. Post-ionophore treatment, PNA binding was still

observable for 92.6 ± 0.6% (mean ± SEM; *n* = 3 different males) of sperm from *Arrdc5*−/− mice which was significantly different compared to the 40.6 ± 3.0% (mean ± SEM; *n* = 3 different males) of sperm from *Arrdc5*+/+ littermates that still had detectable PNA binding thus signifying that >50% had undergone the acrosome reaction (Fig. 4g, h).

Next, we compared the IVF capacity of sperm from *Arrdc5*−/− and *Arrdc5*+/+ littermates. In standard conditions with denuded oocytes possessing intact zona pellucidas, the day 1 two-cell cleavage rate for sperm from *Arrdc5*+/+ males was within the normal range at 55.1 ± 10% (mean ± SEM; *n* = 3 different mice), whereas sperm from *Arrdc5*−/− males (*n* = 3 different mice) yielded no (0%) two-cell embryos (Fig. 4i, j) and sperm penetration of the zona pellucida for oocyte binding was not evident at 5 h of incubation (Fig. 4i). To test whether any sperm from *Arrdc5*−/− males could fuse with the oocyte plasma membrane and achieve fertilization we conducted IVF with ovulated oocytes that had the zona pellucida removed. Although significantly reduced by 3.6-fold compared to sperm from *Arrdc5*+/+ control littermates, sperm from *Arrdc5*−/− males yielded a 23.8 ± 9.2% (mean ± SEM; *n* = 4 different mice) day 1 two-cell cleavage rate (Fig. S13a). Also, 8.5 ± 5.1% of zona pellucida-free oocytes fertilized by sperm from *Arrdc5*−/− males were able to advance to blastocysts which was reduced significantly by 5.5-fold compared to sperm from *Arrdc5*+/+ littermates (46.6 ± 10.6% blastocyst rate) in the same IVF conditions (Fig. S13b, c). Finally, we examined the efficiency of sperm nuclei from *Arrdc5*−/− males to fertilize oocytes and produce viable embryos when introduced directly by intracytoplasmic sperm injection (ICSI). Sperm from *Arrdc5*−/− males introduced by ICSI resulted in a cleavage rate of ~34% and in vitro blastocyst rate of ~4%, which was reduced by 1.7-fold and 7.5-fold compared to the ~57% cleavage rate and ~30% blastocyst rate generated by ICSI with sperm from *Arrdc5*+/+ males, respectively (Fig. S13a–c). Taken together, these findings demonstrate that sperm produced in the absence of ARRDC5 are incapable of final maturation and achieving natural fertilization, but some retain the capacity for fusion with the oocyte plasma membrane and egg activation when barriers are removed.

## Spermatocytogenesis is not disrupted in ARRDC5-deficient mice

Spermatogenesis can be defined broadly by three major processes; spermatocytogenesis, which is the activities of diploid undifferentiated and differentiating spermatogonia that yields meiotic spermatocytes, meiosis during which unique haploid genomes are created through chromosome recombination and DNA reducing cell divisions, and spermiogenesis which is the elaborate transformation of round haploid spermatids that yields specialized elongated spermatids. In mammals, these processes occur over successive cycles of the seminiferous epithelium; defined as 12 stages in mice[27]. To further examine the underlying cause of multifaceted sperm parameter deficiencies in *Arrdc5*−/− mice, testes from adult males (2–4 months old) were processed for histopathological analysis of the seminiferous cycle. The

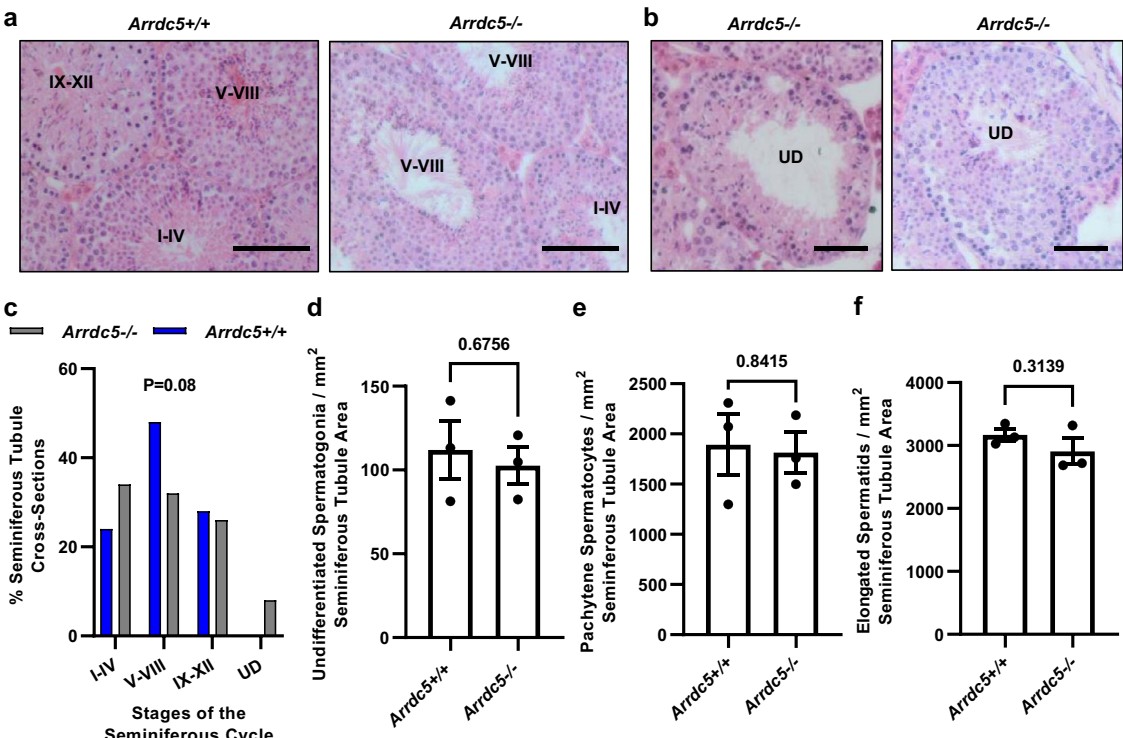

**Fig. 5 | Assessment of spermatocytogenesis in *Arrdc5*−/− mice. a** Seminiferous tubule cross-sections from testes of adult *Arrdc5*+/+ and *Arrdc5*−/− littermates. Tubules at different groupings for the stage of the seminiferous cycle are indicated. Bars are 50 μm. **b** Seminiferous tubule cross-sections from testes of adult *Arrdc5*−/− mice for which a stage of the seminiferous cycle was not definable (indicated at undefined or UD). Bars are 50 μm. **c** Quantitative comparison of distribution for the 12 stages of the seminiferous cycle in cross-sections of testes from adult *Arrdc5*+/+ and *Arrdc5*−/− littermates, grouped as I–IV, V–VIII, and IX–XII. Data bars are mean for at least 50 cross-sections from *n* = 3 different animals of each genotype. Cross-sections for which a clearly definable stage could not be assigned (undefined or UD) were observed from testes of *Arrdc5*−/− mice only. **d**–**f** Quantitative comparison of undifferentiated spermatogonia **d**, pachytene spermatocytes **e**, and round spermatids **f** in cross-sections of testes from adult *Arrdc5*+/+ and *Arrdc5*−/− littermates. Data bars are mean ± SEM and dots represent values for individual animals (*n* = 3 for each genotype and 150 cross-sections). Images in **a** and **b** are representative of three independently repeated experiments. For quantitative comparisons, differences between genotypes were analyzed statistically using unpaired two-tailed *t*-tests. Source data are provided as a Source Data file.

12 stages of the cycle were broadly assessed as groupings of I–IV, V–VIII, and IX–XII. Although cross-sections of seminiferous tubules in all stages of the cycle groupings could be observed for testes from both *Arrdc5*+/+ and *Arrdc5*−/− mice (Fig. 5a), 8.3 ± 2.1% (mean ± SEM; *n* = 3 different males and 50 cross-sections) of tubules in cross-sections from *Arrdc5*−/− testes appeared to be disorganized and a logical stage of the seminiferous cycle group could not be assigned (Fig. 5b). As such, from assessment of numerous cross-sections, distribution across the stage of the cycle groupings was found to be misaligned between testes from *Arrdc5*+/+ and *Arrdc5*−/− males with a reduction of frequency for group V–VIII tubules in testes from *Arrdc5*−/− males (Fig. 5c); however, the overall alignment was not statistically different between genotypes (*P* = 0.08). Next, we quantified the number of undifferentiated spermatogonia, pachytene spermatocytes, and round spermatids in cross-sections of testes and found no differences between *Arrdc5*−/− (*n* = 3 different males and 50 cross-sections) and *Arrdc5*+/+ (*n* = 3 different males and 50 cross-sections) littermates (Fig. 5d–f). These results suggest that two of the three major phases of spermatogenesis— amplification of spermatogonia via mitotic divisions and two meiotic divisions of spermatocytes that produces haploid round spermatids—are not grossly disrupted in the absence of ARRDC5 function. In addition, these findings imply that spermatogonial stem cell capacity to support the foundation for steady-state spermatogenesis is not impaired by ARRDC5 deficiency.

### Defective spermiogenesis in mice lacking ARRDC5
Considering that spermatogonial and spermatocyte populations are not compromised in *Arrdc5*−/− male mice and ARRDC5 protein expression coincides with haploid spermatids, we next focused on assessing the process of spermiogenesis to further define the cause of poor sperm parameters and sterility. Spermiogenesis is the penultimate step in the generation of spermatozoa, a process in which round haploid spermatids transform to a specialized elongated state (Fig. 6a). Based on 3D scanning electron microscopy (SEM), all epididymal sperm from *Arrdc5*−/− mice possessed head abnormalities that were strikingly different than the normal crescent-shaped morphology of sperm from *Arrdc5*+/+ mice (Fig. 6b). These included enlarged heads, malformed acrosomes, misshapen equatorial segments, lack of hook rims, flattened heads, and retained cytoplasm (Fig. 6b). In addition to head abnormalities, transmission electron microscopy (TEM) imaging revealed defects in the neck, midpiece, and tail of sperm from *Arrdc5*−/− mice. Notably, we observed sperm lacking the post acrosomal segment, disorganized mitochondrial sheath, retained cytoplasm, and multiple axonemes (Figs. 6c and S14). Furthermore, outcomes of Comet assay analysis revealed significant DNA damage in epididymal sperm from *Arrdc5*−/− mice compared to sperm from *Arrdc5*+/+ littermates (Fig. 6d, e). Collectively, these findings indicate that ARRDC5 deficiency during spermatogenesis disrupts critical steps in the final phase of differentiation including normal head morphogenesis, cytoskeletal reorganization, mitochondrial positioning, and flagellar development.

The final step in spermiogenesis, referred to as spermiation, is the release of fully elongated spermatids from supporting somatic Sertoli cells into the seminiferous tubule lumen for transport to the epididymis[12]. This process involves aligning of elongated spermatids at the luminal edge of the epithelium, shedding of residual cytoplasm,

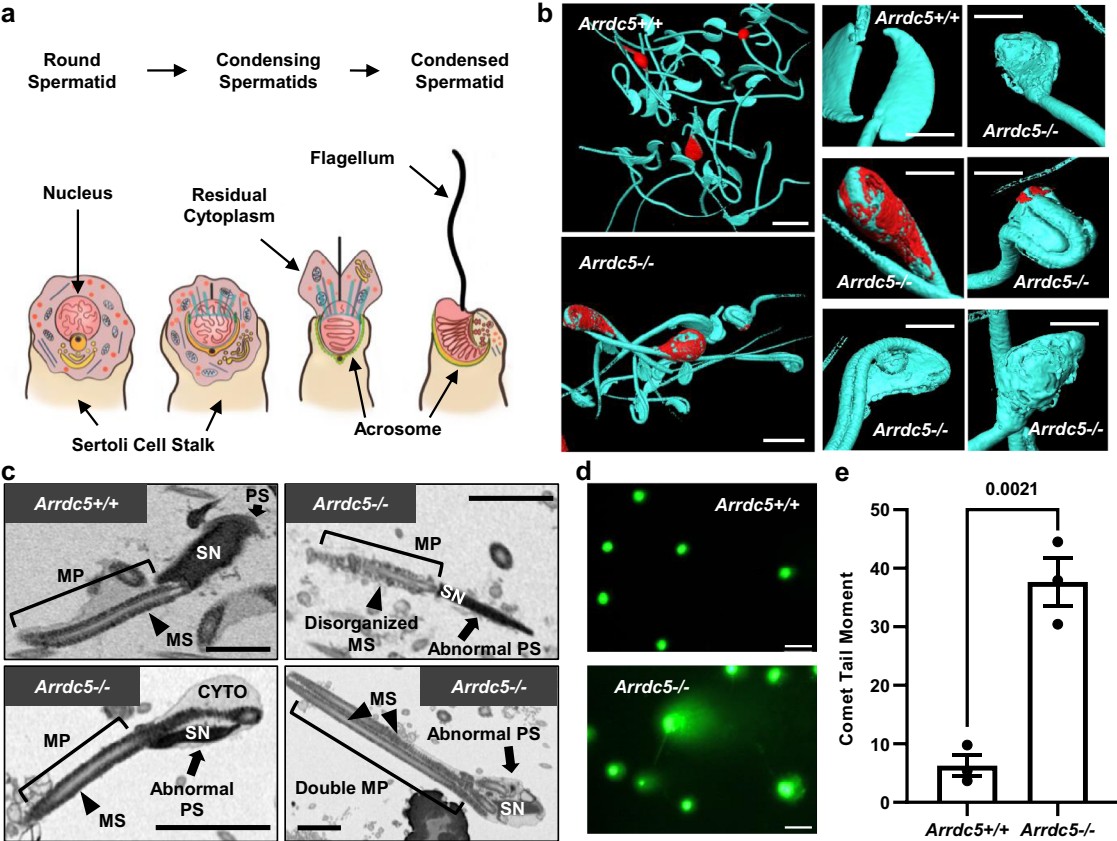

**Fig. 6 | Assessment of spermiogenesis in *Arrdc5*−/− mice. a** Schematic depicting the major steps in spermiogenesis that occur in the seminiferous epithelium of mouse testes. **b** Three-dimensional scanning electron microscopy for epididymal sperm of adult *Arrdc5*+/+ and *Arrdc5*−/− littermates. A variety of sperm head, neck, midpiece, and tail defects were observed for sperm of *Arrdc5*−/− mice, including retained cytoplasm as indicated by red coloring. Bars are 5 µm. **c** Transmission electron microscopy for epididymal sperm of adult *Arrdc5*+/+ and *Arrdc5*−/− littermates. Defects in the sperm nucleus (SN), post acrosomal segment (PS, arrows), and mitochondrial sheath (MS, arrowheads) of the midpiece (MP, brackets) were observed for sperm of *Arrdc5*−/− mice compared to sperm from *Arrdc5*+/+

littermates. Bars are 5 µm. **d** Comet assay analysis for DNA fragmentation of epididymal sperm from adult *Arrdc5*+/+ and *Arrdc5*−/− littermates. Bars are 1 µm. **e** Quantitative comparison of DNA fragmentation levels (defined by the Comet tail moment) for epididymal sperm from adult *Arrdc5*+/+ and *Arrdc5*−/− littermates. Data bars are mean ± SEM and dots represent values for individual animals ($n = 3$ different animals of each genotype and ten images of 600−700 sperm). Images in **b**–**d** are representative of three independently repeated experiments. Differences between genotypes were analyzed statistically using unpaired two-tailed *t*-tests. Source data are provided as a Source Data file.

dissolution of intercellular bridges that connect cohorts of the haploid cells, and disassembly of ectoplasmic specializations that attach the elongated spermatid head to Sertoli cells[28]. Disruption of these steps can result in low sperm numbers in the epididymis, abnormal morphology of the sperm, and sloughing of interconnected sperm. Interestingly, in epididymal flushing from *Arrdc5*−/− males, we observed clumps of malformed sperm in which multiple heads and tails were apparent (Fig. 7a). These conglomerated immature sperm could also be detected by SEM and appeared to be bound by residual cytoplasm (Fig. 7b). In addition, interconnected sperm nuclei could be observed by TEM (Figs. 7c and S14). None of these oddities were observed for any epididymal sperm samples from *Arrdc5*+/+ littermates. Lastly, we explored whether the timing of elongated spermatid release is disrupted in *Arrdc5*−/− mice. Normally, in the mouse, fully formed step 16 spermatids are released from Sertoli cells via spermiation at stage VIII of the seminiferous cycle when they are aligned at the luminal edge[29]. In cross-sections of testes from *Arrdc5*−/− mice, we observed apparent step 16 elongated spermatids mispositioned deep in the epithelium of 20% of tubule cross-sections (Fig. 7d), an oddity that was not observed in any cross-sections of testes from *Arrdc5*+/+ littermates. Together, these results suggest that ARRDC5 function during spermiogenesis is important for critical steps in the spermiation process.

## ARRDC5 regulation of spermatogenesis is intrinsic to germ cells

Finally, we aimed to determine whether defects in sperm production of *Arrdc5*−/− mice are intrinsic to germ cells. Although the expression profile for *Arrdc5* mRNA and protein suggested germ cell specificity in adulthood, the possibility of genetic inactivation causing impairments in somatic cell function to properly support normal spermatogenesis could not be ruled out by studying the knockout mouse only. To confirm the indication, we isolated spermatogonial stem cells (SSCs) from adult *Arrdc5*−/− mice and transplanted them into testes of *Nanos2* knockout males that are ablated of endogenous germline but have functional *Arrdc5* alleles and testicular somatic cells capable of supporting donor-derived spermatogenesis[30]. Donor *Arrdc5*−/− cells were transplanted into one testis of each recipient male and the contralateral testis was transplanted with cells isolated from *Arrdc5*+/+ littermates (Fig. 8a). At 3 months post-transplantation, recipient testes and epididymides were collected for assessment of donor-derived spermatogenesis. Sperm with normal morphology were observed in epididymal flushing from testes transplanted with *Arrdc5*+/+ SSCs, whereas sperm with abnormal morphology were observed in flushing from epididymis associated with testes transplanted with *Arrdc5*−/− SSCs (Fig. 8b). In addition, seminiferous tubules with normal appearing epithelium and complete spermatogenesis were observed in cross-sections of recipient testes transplanted with *Arrdc5*+/+ SSCs, but the

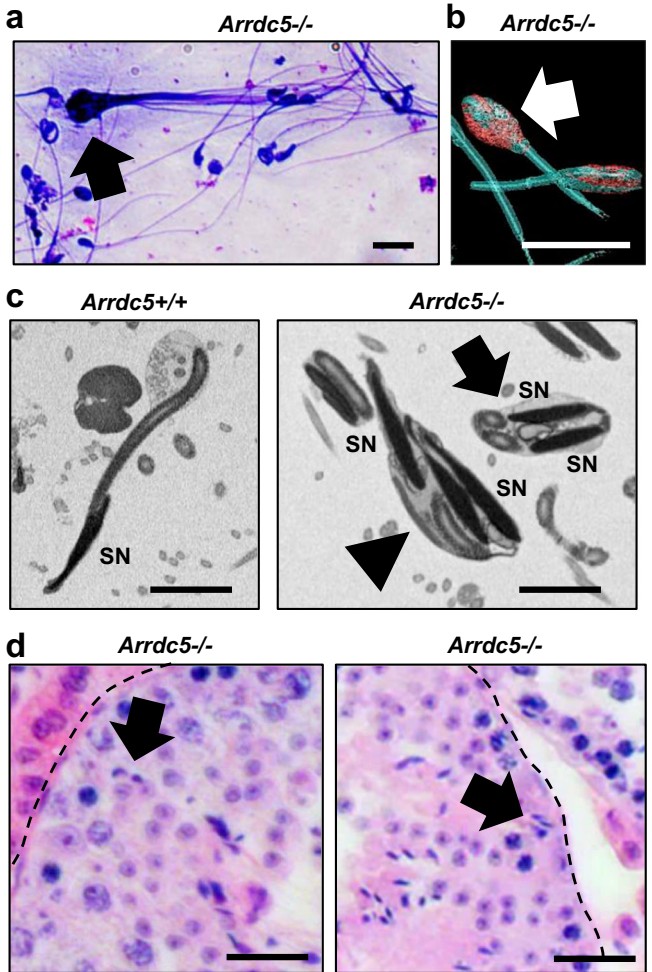

**Fig. 7 | Assessment of spermiation defects in *Arrdc5*−/− mice. a** Dip Quick stained epididymal sperm from adult *Arrdc5*−/− mice. Arrow indicates a conglomerate of sperm that have been prematurely released from the seminiferous epithelium. Bar is 10 μm. **b** Three-dimenional scanning electron microscopy for epididymal sperm from adult *Arrdc5*−/− mice with an enlarged head wrapped in residual cytoplasm (arrow). Bar is 10 μm. **c** Transmission electron microscopy for epididymal sperm from adult *Arrdc5*+/+ and *Arrdc5*−/− littermates. Sperm heads with multiple nuclei (SN) that are interconnected (arrow) or enfolded by cytoplasm (arrowhead) were observed for *Arrdc5*−/− mice, an oddity that is not observed with sperm from *Arrdc5*+/+ mice. Bars are 5 μm. **d** Seminiferous tubule cross-sections from testes of adult *Arrdc5*−/− mice in which elongated spermatids (indicated by arrows) are embedded deep in the seminiferous epithelium. Bars are 25 μm and dashed lines indicate seminiferous tubule basement membrane. All images are representative of three independently repeated experiments.

epithelium in tubules colonized by *Arrdc5*−/− SSCs appeared disrupted (Fig. 8b). Collectively, these findings demonstrate that ARRDC5 function in spermatogenesis is intrinsic to germ cells.

## Discussion

Arrestin family molecules are well known to be highly conserved throughout eukaryotic evolution and have been characterized in an array of mammalian cell types[17,20,22], yet expression and functional importance in the testis and spermatogenesis of any species have been undefined. A previous study found enriched expression of ARRDC4 in the mouse epididymis and genetic inactivation led to a reduction in fertility[23], likely through extrinsic influences that the organ has on the final maturation of sperm to gain fertilizing capacity. Aside from this single study, a characterized expression for any other α-arrestin molecule has, at present, not been reported in the peer-reviewed

scientific literature. In addition, the functional relevance of any α-arrestin molecule in regulating spermatogenesis is currently unknown. Thus, the results of the present study represent a breakthrough discovery of ARRDC5 as an α-arrestin that is likely expressed exclusively in germ cells across mammalian species and has an essential role in normal sperm morphogenesis and male fertility. This finding opens many new avenues for investigation, including the linkage of mutations to infertility in an array of mammalian males and the development of a novel male contraceptive approach. Although our findings clearly show that ARRDC5 is an essential regulator of sperm morphogenesis, the mechanism of action is undefined and filling this gap in knowledge will be important for understanding how genetic deficiency could lead to infertility as well as targeting the molecule for male contraceptive development.

In somatic cells, arrestin molecules are known to function as modulators of ubiquitination which is a three-step posttranslational modification process that covalently attaches ubiquitin molecules to proteins[22]. A variety of cellular functions are influenced by this process, including tagging of proteins for degradation, protein trafficking, and shaping of chromatin structure[31]. Following activation of the ubiquitin molecule by an E1 enzyme, transfer to the protein is facilitated by an E2 conjugating enzyme, and finally, an E3 ligase creates the covalent bond. While the E1 and E2 enzymes are universal in their functions, the family of E3 ligases associates with arrestin molecules that serve as adapters to determine substrate specificity for tagging of targeted proteins[20–22]. In mammals, three major classes of E3 ubiquitin ligases are expressed: HECT type, RING-type, and RBR-type. Although several previous studies have profiled the expression of genes involved with ubiquitination in mammalian testes[32–35], identification of functional roles in regulating spermatogenesis have been limited. In mice that are homozygous for a null allele encoding the RING-type E3 ligase RNF8, spermatogenesis is defective with several impairments that reflect OAT, including reduced sperm count and high percentages of immotile and morphologically abnormal sperm[36]. In addition, although male mice that are knockout for the RING-type E3 ligase RLIM (also known as RNF12) are fertile, conditional inactivation of the gene in germ cells results in a reduction of sperm numbers, motility, and fertilization capacity[37]. Beyond spermatogenesis, expression of both RNF8 and RLIM occurs in an array of tissues and cell types where the molecules have important biological roles. Both male and female RNF8 null mice display a range of developmental abnormalities that reduce lifespan and increase incidences of cancer[38–40]. For RLIM, due to being X-linked and having a role in imprinting, female embryos that are homozygous for null alleles die during the peri-implantation period and deficiencies in expression and/or function are linked with multiple disease states[41]. Recently, the RING-type E3 ubiquitin ligase RNF133 was identified as having testis-enriched expression in mice and genetic inactivation was shown to cause subfertility due to impaired sperm morphogenesis and motility[42]. Like it's α-arrestin family members, whether ARRDC5 interacts with any of the E3 ubiquitin ligase subtypes to facilitate substrate specificity remains to be explored experimentally.

In general, the ARRDC subfamily of α-arrestins influence ubiquitination via association with HECT-type E3 ubiquitin ligases through a PPxY (or PY) motif; notably, ARRDC5 is unique in that it is the only family member that lacks this feature[20,22]. Thus, if ARRDC5 does modulate ubiquitination in male germ cells, it will likely be via a novel mechanism that involves association with RING-type or RBR-type E3 ligases. Intriguingly, our finding of testis-specific expression for ARRDC5 opens the possibility of how spermatogenic lineage-specific protein ubiquitination could have evolved in Mammalia.

In addition to facilitating ubiquitination, arrestin molecules are known to function as modulators of G protein couple receptor (GPCR) signaling in somatic cells[43]. The visual/β-arrestins are well characterized for having roles in regulating the trafficking and inactivation of

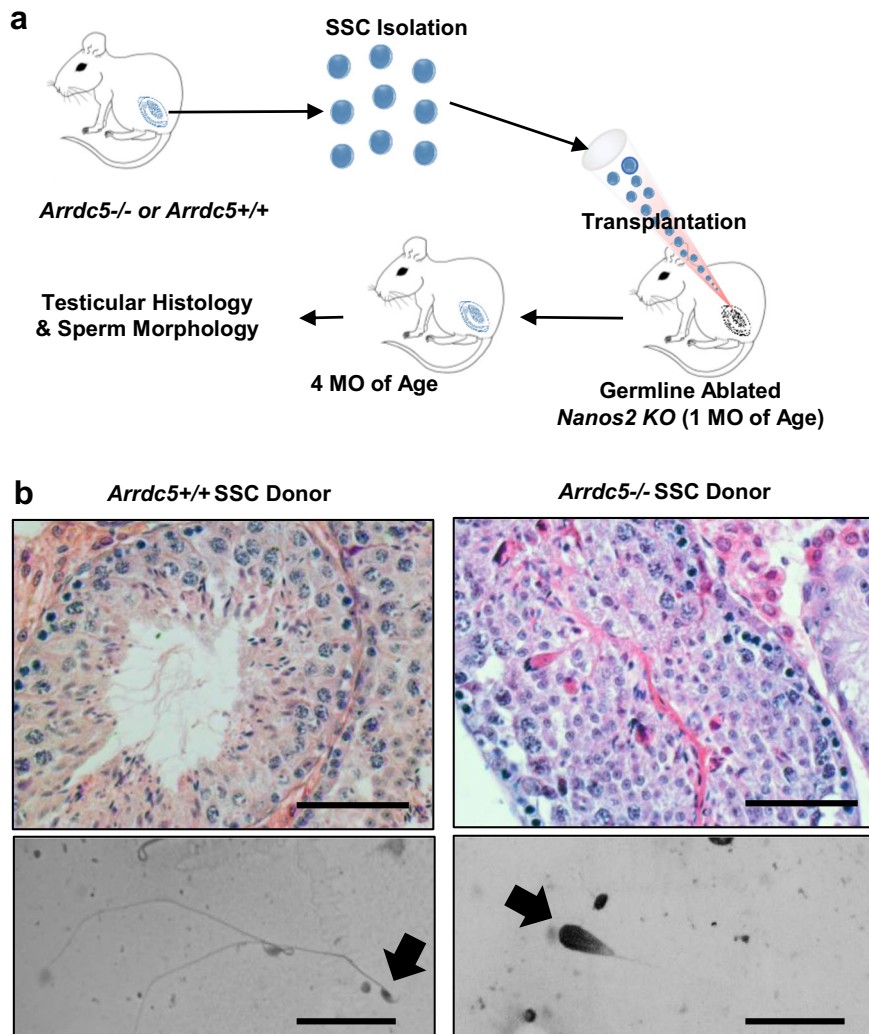

**Fig. 8 | Assessment of germ cell-intrinsic defects of *Arrdc5−/−* mice. a** Schematic of experimental strategy to assess whether defects in spermiogenesis are intrinsic to germ cells of *Arrdc5−/−* mice. **b** Seminiferous tubule cross-sections from testes (upper panel) and epididymal flushing (lower panel) of recipient mice 3 months after transplantation with spermatogonial stem cells from adult *Arrdc5+/+* or *Arrdc5−/−* littermates. Normal spermatogenesis was observed for recipient males

transplanted with spermatogonial stem cells from *Arrdc5+/+* donor males, whereas abnormal spermatogenesis was observed for recipients transplanted with spermatogonial stem cells from *Arrdc5−/−* donor males. Bars are 25 µm (upper panel of cross-sections) and 50 µm (lower panel of flushing). Images are representative of two independently repeated experiments.

GPCRs through binding interactions at both the N- and C-terminal domains[44,45]. These regions of β-arrestin molecules contain clathrin and AP-2 binding sites that mediate endocytosis of GPCRs. Comparatively, α-arrestins lack β-arrestin N-terminus helix domains as well as clathrin binding sites[17]. Thus, whether α-arrestins function as modulators of GPCRs remains to be elucidated and could be cell type dependent. At present, most of what is known about mechanisms of action for α-arrestins in eukaryotic organisms comes from studies of budding yeast[17,20,22]. In this model system, emerging evidence suggests that α-arrestins function in the endocytosis of nutrient transporters to regulate cellular responses to metabolic signaling[46–48]. In mammalian cells, heterodimerization of β-arrestins and ARRDC1 to modulate GPCRs has been reported[49]. In addition, ARRDC1-4 are known to interact with nutrient transporters in mammalian somatic cells[19,50]. In contrast, although ARRDC5 possesses an arrestin domain, the N- and C-terminal regions are distinct, and its molecular functions in any mammalian cell type are largely unknown[17]. Our discovery that ARRDC5 is expressed uniquely by testicular germ cells across Mammalia and plays a critical role in sperm formation opens an intriguing new area of investigation into evolutionary biology and future

investigations into its mechanism of actions may yield important information for understanding how other ARRDCs function.

Oligoasthenoteratospermia (OAT) is a common clinical presentation for men as well as domestic animals and endangered species that are infertile, yet, at present, the molecular deficiencies that could lead to the disorder are largely undefined. Our discovery that *Arrdc5−/−* mice are infertile due to severe OAT, have an otherwise normal physiology, and the gene is highly conserved as a 1:1 ortholog across mammalian species opens the intriguing possibility that deleterious mutations could be underlying causes of fertility defects. To fertilize an oocyte and yield a developmentally competent embryo either in vivo or in vitro, sperm must possess a properly formed head, including a correctly assembled haploid genome and intact acrosome, and a flagellum that can hyperactivate. These attributes are created during the final phase of spermatogenesis via the intricate process of spermiogenesis. Inside the female reproductive tract, sperm must undergo capacitation which involves hyperactivation of the flagellum to propel the sperm to the site of fertilization and acrosomal exocytosis, which is required for penetration of the zona pellucida and progression of sperm-egg fusion. We found that sperm generated in

the absence of ARRDC5 function are unable to hyperactivate and possess malformed acrosomes; thus, neither in vivo nor in vitro fertilization of ovulated eggs that possess intact zona pellucida is possible. However, we found that some sperm from *Arrdc5*−/− mice can fertilize oocytes that have had the zona pellucida removed or have been microinjected and these embryos advance to the blastocyst stage in vitro, albeit at a low frequency. These findings suggest that some sperm generated in the absence of ARRDC5 function possess a normal haploid genome, but whether embryos formed in this manner are competent for gastrulation and can yield live offspring is unknown. Addressing this gap in knowledge is likely important for assisted reproductive technology settings in which males present with OAT, and either IVF or intracytoplasmic sperm injection (ICSI) will be used to generate embryos for the intent of establishing pregnancies. If the male possesses deleterious mutations in the *Arrdc5* gene that lead to the formation of sperm with an improperly packaged paternal genome, use in assisted reproductive technologies would be questionable.

Over the last two decades, several studies have described various genetic deficiencies in mouse models that lead to poor sperm parameters[5,51–57]. Although many of the phenotypes include aspects of OAT, most models do not fully reflect the pathologies of infertile men, domestic animals, or endangered species and expression of the genes occurs in multiple organs as well as testes. Although several genes with testis-enriched expression have been discovered in mice to have important roles in spermatogenesis, knowledge of genes expressed specifically in testicular germ cells that have an essential intrinsic role and functional translation from discovery in mice to evolutionary conservation in other mammalian species has been limited. Studies by Zheng et al., 2007, discovered that the gene spermatid maturation 1 (*Spem1*) is testis-specific in mice and humans, and inactivation in mice leads to defective shedding of cytoplasm during spermiogenesis, as such, the sperm heads are malformed and positioned incorrectly for fertilization, thus leading to infertility. However, the epididymal sperm concentration of *Spem1*−/− mice was in the normal range and defects in the neck, midpiece, and flagella were not observed[57]. Similarly, expression of coiled-coil domain-containing 62 (*Ccdc62*) is specific for testes of mice and knockout causes sterility due to severe malformations of the sperm head and defective flagella, but epididymal sperm count is not altered[58]. Notably, two genes have been discovered in mice that are expressed exclusively in testes and strongly reflect OAT; capping protein (actin filament) muscle Z-line, alpha (*Capza3*)[59], and homeodomain-interacting protein kinase 4 (*Hipk4*)[55]. Studies by Geyer et al., 2019, found that an inactivation point mutation in the *Capza3* coding sequence was the cause of infertility in mice possessing an ENU-induced mutation that had been referred to as *repro32*. Expression of CAPZA3 is testis-specific and mice homozygous for the inactivating *repro32* mutation have low epididymal sperm numbers, poor sperm motility, and abnormal sperm morphology[59]. With HIPK4, expression is predominant, but not exclusive, to the testis of mice and humans (GTEx Portal), and genetic knockout leads to infertility due to reduced sperm production, impaired sperm motility, and abnormal head as well as tail morphogenesis[55]. The morphological abnormalities of sperm produced in the absence of CAPZA3, HIPK4, or ARRDC5 are similar but whether these molecules are connected mechanistically during normal spermiogenesis is unknown and should be an interest of future investigations. Interestingly, deficiency in ARRDC5 appears to have a wider range of defects than either CAPZA3 or HIPK4 deficiency. For example, macrospermia was observed with *Arrdc5*−/− mice but this abnormality was not reported for *Capza3*−/− or *Hipk4*−/− mice. Also, sperm capacitation was reported to be normal for sperm of *Hipk4*−/−[55], but is significantly impaired for sperm from *Arrdc5*−/− mice. Clearly, *Arrdc5*−/− mice offer a unique model of study for better understanding the molecular mechanisms that drive sperm morphogenesis and defining how deficiencies can lead to OAT in males of a variety of mammalian species.

Although the exact molecular function of ARRDC5 in male germ cells remains to be elucidated, the presumed biological role as an E3 ubiquitin ligase adapter may make it a druggable target for male contraceptive development. Ideally, a drug-based reversible male contraceptive would target the end phase of spermatogenesis to render sperm incapable of natural fertilization. In this way, the testicular size would not be reduced, the endocrine system would be unperturbed, and precursor spermatogenic cell types (e.g., spermatogonia) would not be impacted. In mice, we found that *Arrdc5* mRNA is detectable in spermatogonia, spermatocytes, and spermatids, but the protein is present in round and elongated spermatids only. In addition, the inferred subcellular localization of the ARRDC5 protein is cytoplasmic. Furthermore, the potential for drug interference of ARRDC5 function to impact physiological systems other than spermatogenesis is potentially minimal, given that *Arrdc5* knockout male and female mice are normal. For these reasons, targeting ARRDC5 with small-molecule inhibitors may provide an excellent avenue for novel male contraceptive development.

## Methods

### Animals
Animal procedures were approved by the Washington State University Animal Care and Use Committee (IACUC). Mouse lines were procured from Charles River Laboratories (CD1/ICR, stock no. 022), The Jackson Laboratory (C57BL/6J, stock no. 000664, and 129S1/svlmJ, stock no. 002448), Ervigo (ICR/CD1, stock no. Hsd-ICR CD-1®) or created at Washington State University. Cattle were prepubertal Holstein bull calves obtained from the Washington State University Knott Dairy. Pigs were prepubertal mixed breed males obtained from a local swine producer.

### Single-cell RNA-sequencing and gene ontology analysis
Testes from prepubertal Holstein breed cattle and mixed breed pigs were collected by castration. For all samples, testicular single-cell suspensions were generated using the methodology previously described by refs. 30,60. Briefly, testicular parenchyma was enzymatically digested with trypsin-EDTA solution (Sigma-Aldrich) followed by Collagenase IV (Worthington) with DNase I (Sigma-Aldrich) using gentleMACS Octo-Dissociator (MiltenyiBiotech). Enzymatic activity was then blocked (10% FBS; Gibco), live cells (~8,000 per sample) were then loaded into a Chromium Controller (10X-Genomics, Inc.), and cDNA libraries were generated using v2 chemistry kits (10X-Genomics, Inc.). Libraries were pooled at proportions netting equal read depth and sequenced in a single lane on an Illumina HiSeq 4000 (Genomics and Cell Characterization Core Facility, University of Oregon). Raw base call files were demultiplexed using the 10X-Genomics Cell Ranger pipeline (v2.1.0) and aligned to the bovine genome (ARS-UCD1.2) or porcine genome (Sscrofa11.1). Raw mice P6 testis library was generated in previous studies[24] and aligned to mouse genome assembly (GRCm39).

A total of seven raw testis transcriptomes (mouse $n = 1$, pig $n = 3$, and cattle $n = 3$) were merged and integrated with R-Studio (v.3.4.4) with Seurat (v2.3.2) and Monocole (v2.6.4) packages. Low-quality cell transcriptomes and doublets were determined and excluded within each library, and gene annotations were standardized between species. The full dataset was then normalized, scaled, and dimensionally reduced using Seurat (v2.3.2). Standard log-normalization preprocessing and identification of variable features were performed for each dataset, followed by placing anchors. The integrated matrix was then used for downstream analysis and clustering visualization by UMAP. Clusters were assigned cell type identities based on the expression of canonical biomarker genes (i.e.,: *Ddx4* for germ cells). Differential gene expression for the germ cell cluster compared to other cell clusters was determined based on average expression and dispersion. Ontology analysis for differentially expressed germ cell genes was

performed using PANTHER (v13.1). Venn diagram analysis was accomplished using Venn Diagram software (https://bioinformatics. psb.ugent.be/webtools/Venn/; Bioinformatics & Evolutionary Genomics).

### RT-PCR analysis

Total cellular RNA was isolated from testicular parenchyma, brain, kidney, liver, heart, adipose, pancreas, lung, muscle, epididymis, and skin of prepubertal and mature cattle, pigs, and mice (Trizol; Invitrogen) and genomic DNA contamination was removed (DNA-free™ Kit, Ambion). For each sample, 1 μg of RNA was reverse transcribed to cDNA using the SuperScript IV First-Strand Synthesis System (Invitrogen). Reactions performed without reverse transcriptase enzyme served as -RT controls, and residual RNA was eliminated (RNaseH, Invitrogen). For *Arrdc5* RT-PCR, primers were designed based on reference genomes for bovine (NM_001128508.2), porcine (KF589203.1), and murine (NM_029799.1). Nucleotide sequences for all primers are listed in Supplementary Table 1. All PCR reactions were performed using Platinum-SuperFi II Green mix (Invitrogen) and amplicons visualized by agarose gel electrophoresis.

### Generation of *Arrdc5*−/− mice

To engineer an inactivated *Arrdc5* allele in the mouse genome, CRISPR-Cas9 gene editing was employed. Dual single guide RNAs (sgRNAs) were designed using CRISPOR[61] and based on the mouse reference genome sequence (NM_029799.1) to delete a large portion of exon 1 of the *Arrdc5* coding sequence (CDS). The top-ranking sgRNAs (CDS_target scores of >80) were selected and synthetically generated (Synthego Inc.) (Supplementary Table 1). To generate ribonucleoprotein (RNP) complexes, sgRNAs and TrueCut™ Cas9 Protein v2 (Thermo Fisher) were diluted at a 1:1 mass ratio.

Next, superovulated 5–7-week-old C57BL/6 J females were paired overnight with 129SvlmJ males. Zygotes were incubated with RNP complexes followed by electroporation (30 V and 3 ms/pulse X 7; BTX, Harvard-Apparatus) as described previously. Following electroporation, zygotes were cultured in EmbryoMax-KSOM (MilliporeSigma) at 37 °C in an atmosphere of 5% $CO_2$ for 3.5 days to produce blastocysts and transferred to synchronized recipients by the non-surgical procedure[62]. Several E0 offspring were produced from these embryo transfers, and a male heterozygous for an *Arrdc5* allele with a 308 bp deletion was selected as a founder to generate an experimental line. Breeding of the $Arrdc5^{308\Delta/+}$ founder male to wild-type C57BL6/J females resulted in germline transmission, and the N1 offspring were then crossed for filial breeding to create an experimental line for analysis. Homozygous $Arrdc5^{308\Delta/\Delta}$ mice were born at expected Mendelian ratios and were viable from birth through adulthood. For genotyping analysis, primers were used to amplify the murine Arrdc5 CDS (Supplementary Table 1). PCR products were visualized by agarose gel electrophoresis and Senger sequencing (BigDye Terminator v3.1 Cycle Sequencing Kit; Thermo Fisher). Mutation detection was made by aligning base calls to the reference genome sequence for murine *Arrdc5* (Accession: ENSMUSG00000073380).

### Generation of *Arrdc5-eGfp* mice

To balance position and off-target potential, two Cas9 target sites were selected using CRISPOR[61]. The sgRNAs (Supplementary Table 1) were generated using a cloning-free method[63], with oligonucleotides (Eurofins-Genomics), in vitro transcription (Thermo Fisher), and column purification (Zymo-Research). To determine the editing potential in mouse zygotes, RNPs of the sgRNAs and Cas9 protein were electroporated into zygotes[62]. At the blastocyst stage, embryos were collected, and editing at the *Arrdc5* target site was analyzed by heteroduplex mobility shift assay[63], using the LF2 and RR2 primers (Supplementary Table 1). The repair template was generated via NEBuilder-HiFi assembly (NEB) in pBluescript SK(+) (Agilent, plasmid #X52328). *eGfp* was amplified from pcDNA3-eGfp (a gift from Doug Golenbock; Addgene plasmid #13031), and 1 kb homology arms were amplified from C57BL6/J genomic DNA. The P2A peptide sequence was included as an extension in the oligonucleotide that overlapped the left homology arm and EGFP. After correct assembly was confirmed by sequencing, a biotinylated dsDNA donor was generated using biotinylated primers (Eurofins-Genomics) and Q5 DNA polymerase. The Cas9-streptavidin fusion (Cas9mSA) was transcribed from PCS2 + Cas9mSA (a gift from Janet Rossant, Addgene plasmid #103882) using the mMessage machine SP6 capped RNA transcription kit (Thermo Fisher) and purified via LiCl precipitation and a subsequent RNA column purification (Zymo-Research). For initial testing of knock-in efficiency, the guide pair, Cas9mSA, and biotinylated donor were co-injected in 2-cell embryos. At the blastocyst stage, the knock-in rate was assessed using nested PCR with the locus primer (LF1) and the other primer within *eGfp* (LR) (Table S1). A >30% knock-in rate was achieved, sufficiently high to generate knock-in animals.

Microinjection of two-cell embryos was performed using the methodology described previously in ref. 64. Briefly, superovulated 5–7-week-old C57BL/6 J females were paired overnight with males of the same strain. Two-cell embryos were recovered by flushing the oviducts with M2 medium (MilliporeSigma). For microinjection, embryos were equilibrated and washed in KSOM-Advanced (MilliporeSigma). Roughly 3–5 pl of a solution containing 50 ng/ml sgRNA, 100 ng/ml Cas9MSA mRNA, and 30 ng/ml dsDNA donor was injected into the nucleus of each blastomere using Sutter/Xenoworks (Novato) micromanipulators. Following microinjection, embryos were cultured to the blastocyst stage in KSOM-AA (CytoSpring) at 37 °C and 5% $CO_2$ and were individually screened to assess the correct knock-in rate. Additional embryos were microinjected and surgically transferred to the oviducts of 0.5 dpc pseudopregnant females. Tail biopsies were obtained from the resulting founder pups and screened by PCR. Two putative founders carrying the correct modification were bred with wild-type C57BL6/J mates, and positive N1 generation offspring were used for further breeding. Some breeders were back-crossed to wild-type C57BL6/J mice for two to three additional generations to further dilute any potential off-target genomic editing.

### Testis histology and immunostaining

Following euthanasia, testicular and epididymal tissues were fixed in Bouin's solution (Sigma-Aldrich Inc.) or 4% paraformaldehyde (PFA; Pierce), respectively, for 24 h at 4 °C, followed by dehydration in graded series of ethanol and xylenes and then embedded in paraffin. Cross-sections of 5 μm were generated and deparaffinized. For testicular and epididymal histology, cross-sections were stained with hematoxylin (Sigma-Aldrich) and eosin (Ricca Chemical) and viewed by light microscopy (IX51; Olympus). Digital images were captured at 100 to 200X magnification (TH4-100 digital camera and CellSens Dimensions software v.3.2; Olympus).

For fluorescent immunostaining analysis, tissue cross-sections were processed for antigen retrieval by incubation. Nonspecific antibody binding was blocked by incubating sections in a solution containing 10% donkey or normal goat serum, 1% BSA (Sigma-Aldrich), and 0.025% Triton X-100 (Sigma-Aldrich) in PBS at room temperature for 2 h. Cross-sections were then incubated overnight at 4 °C with primary antibody or normal IgG (control) in a solution containing 1% BSA (Sigma-Aldrich) in 0.025% Triton X-100 (Sigma-Aldrich) in PBS, followed by washing in PBS 3x and then incubation with secondary antibody at room temperature for 2 h. All antibodies and dilutions used are provided in Supplementary Table 1. Coverslips were then mounted with Prolong-Gold reagent containing DAPI (Thermo Fisher) and viewed by fluorescence microscopy (IX51, Olympus). Digital images were captured at 200−400X magnification.

## Fertility assessment, in vitro fertilization, and intracytoplasmic sperm injection

To assess the impacts of ARRDC5 loss-of-function on fertility, *Arrdc5−/−* male and female mice were paired at 2 months of age with pubertal wild-type C57BL6/J counterparts. Controls were male littermates possessing functional *Arrdc5* alleles (*Arrdc5*+/+) that were also paired with C57BL6/J females. All pairings were maintained for 4 mo, and the number of offspring born was recorded as a measure of fecundity.

For in vitro fertilization analysis, the cauda of the epididymis from *Arrdc5−/−* and *Arrdc5*+/+ mice were excised following euthanasia and diced in EmbryoMax-M2 Medium (M2, MilliporeSigma). After 15 min of incubation at 37 °C, sperm suspensions were collected, pelleted by centrifugation (at $600 \times g$ for 7 min), and resuspended in EmbryoMax Human-Tubal-Fluid medium (HTF, MilliporeSigma) containing BSA (3 mg/mL, Sigma-Aldrich) at a concentration of $1 \times 10^7$ cells/ml followed by incubation at 37 °C for 1 h to induce capacitation. Next, cumulus-oocyte complexes (COCs) were harvested from the oviducts of superovulated C57BL6/J female mice and suspended in HEPES-buffered M2 media. Cumulus cells were removed by incubation in a hyaluronidase solution (0.1% w/v hyaluronidase type IV, Sigma-Aldrich) at 37 °C for 15 min. Finally, enzymatic activity was blocked by washing denudated oocytes in TYH + BSA media, and ~20 oocytes were placed per well containing pre-equilibrated HTF + BSA media, and in vitro capacitated sperm were added to each well at a concentration of $1 \times 10^6$ cells/ml followed by incubation at 37 °C for 4 h in a humidified incubator with an atmosphere of 5% $CO_2$ in the air. For zona pellucida-free IVF experiments, denudated oocytes were incubated in Tyrode's acid solution (Sigma-Aldrich Inc.) for 30 s, followed by washing several times in HTF + BSA before the addition of sperm at a concentration of $1 \times 10^5$ cells/ml. After the 4 h incubation period, sperm-oocyte complexes were washed and cultured in pre-equilibrated EmbryoMax-Advanced KSOM media (MilliporeSigma) at 37 °C in a humidified incubator with an atmosphere of 5% $CO_2$ in the air and under mineral oil (Ovoil, Vitrolife). Two-cell cleavage and blastocyst rates were calculated at 24 and 72 h post-onset IVF.

For intracytoplasmic sperm injection (ICSI), 3–6-week-old C57BL6/J x DBA2/J hybrid females were superovulated using intra-peritoneal injection of 5 IU pregnant mare serum gonadotropin (PMSG, Prospec Bio), followed 46–48 h later by injection of 5 IU human chorionic gonadotropin (hCG, Prospec Bio) and metaphase II eggs were collected from ampullae of the oviducts 13–16 h later. Cumulus cells were detached using 3 mg/ml hyaluronidase and washed in M2 medium (Sigma-Aldrich) and placed in a humidified incubator containing 5% $CO_2$ in air at 37 °C until use. Immobilized sperm heads were then isolated by sonication and drawn into a Piezo drill-driven micropipette and injected into the egg cytoplasm with a minimal volume of media. Following sperm head microinjection, eggs were cultured in KSOM (MilliporeSigma) under mineral oil in a 37 °C, 5% $CO_2$ incubator.

## Sperm parameter analysis

Epididymis from adult *Arrdc5−/−* and *Arrdc5*+/+ mice (8–16 weeks of age) were excised following euthanasia, as previously described, in a standard volume of 300 µl M2 medium (MilliporeSigma) followed by incubation for 15 min at 37 °C in a humidified incubator with an atmosphere of 5% $CO_2$ in the air. Sperm concentration and percent motility were calculated using computer-assisted sperm analysis (CASA, Microptic) using SCA software (v1.mouse, Microptic). For morphology analysis, sperm were spread on glass slides, processed for differential Dip Quick staining (JorVet), visualized by light microscopy at 400X or 630X magnification, and the percentage with abnormal head size and shape, midpiece deformities, and tail defects was calculated. For each sample, a minimum of 250 sperm in 10 or more fields of view were scored for normal or abnormal morphology.

## Sperm capacitation analysis

Epididymal sperm were collected, pelleted ($600 \times g$ for 7 min), and incubated for 1 h in HTF + BSA media as previously described. Motility and hyperactivation rates were calculated by CASA (Microptic) at 0 and 1 h of incubation. In vitro acrosome reaction was induced by adding the Ca2+ ionophore A23187 (Sigma-Aldrich) to sperm suspensions at a concentration of 10 mM, followed by incubation for 1 h at 37 °C in a humidified incubator with an atmosphere of 5% $CO_2$ in the air. Next, sperm were pelleted ($600 \times g$ for 7 min), fixed in 4% PFA in PBS for 15 min, spread on glass slides, and permeabilized by incubation for 5 min in a solution of 0.3% Triton X-100 in PBS. To assess acrosome exocytosis, slides were then incubated for 30 min in PBS containing 2% BSA, 0.01% Triton X-100, and fluorescein-conjugated peanut agglutinin (FITC-PNA, 10 µg/ml, Sigma). After washing with PBS, slides and coverslips were mounted with Vectashield-HardSet reagent containing DAPI (Vector-Labs) and then imaged with a DMi8 inverted fluorescent microscope (Leica-Microsystems; LasX v3.3.0) at 63X magnification. For each sample, a minimum of 250 sperm in 10 or more fields of view were scored for the presence or absence of FITC-PNA staining.

## Sperm DNA integrity analysis

The nuclear DNA integrity of sperm was evaluated using a modified alkaline single-cell gel electrophoresis COMET assay described previously in ref. 65. Briefly, glass slides were pre-coated with 1% normal-melting-point agarose (Invitrogen) dissolved in TBE buffer (Sigma-Aldrich). Epididymal sperm collected from adult *Arrdc5*+/+ and *Arrdc5−/−* mice were diluted in 0.75% low-melting-point agarose in TBE at a concentration of $1 \times 10^4$ cells/ml and affixed to pre-coated slides followed by mounting of coverslips and cooling at 4 °C until solidification (~5 min). Coverslips were then gently removed, and slides were covered with a topcoat of 0.75% low-melting-point agarose in TBE. Next, slides were incubated in lysis solution I (100 mM Na2−EDTA, 10 mM Tris, 2.5 M NaCl, pH 11.0, and 20 mg proteinase K; Sigma-Aldrich) for 1 h at 56.8 °C followed by washing with ultrapure water and then incubation in lysis solution II (100 mM Na2−EDTA, 10 mM Tris, 2.5 M NaCl, pH 11.0, 40 mM dithiothreitol, and 2% Triton X-100, Sigma-Aldrich) for 2 h at 4 °C. Slides were washed in ultrapure water and then immersed in cold alkaline electrophoresis solution (300 mM NaOH, 1 mM Na2−EDTA, pH O13.0; Sigma-Aldrich) for 20 min. Electrophoresis was then performed for 25 min at 3 V/cm and 270 mAmp, after which slides were washed in TBE and incubated in increasing concentrations of ethanol solutions (70, 93, and 100%) for 5 min each. Lysed nuclei were stained with SYBR-Safe DNA Gel Stain (Thermo Fisher) diluted in TBE for 15 min and then washed in TBE for 15 min to reduce background staining. Finally, slides were evaluated using fluorescence microscopy (DMi8 inverted microscope; Leica-Microsystems) at 200X magnification. For each slide, 600–700 sperm captured in 10 images were assessed per genotype using Comet-Score software (v1.5, TriTek-Corp).

## Electron microscopy

Epididymal sperm from adult *Arrdc5*+/+ and *Arrdc5−/−* mice were collected in TYH medium (Card Medium, Cosmo-Bio LTD) and centrifuged for 3 min at $500 \times g$. The supernatant was removed, and pellets were resuspended in an aqueous solution of 2% glutaraldehyde (Ted-Pella Inc.), 2% PFA (Electron Microscopy Sciences), 0.1 M cacodylate (Ted-Pella Inc), and 2 mM CaCl2 (Ted-Pella Inc). Samples were fixed for 6 h at room temperature before microwaving at 300 W for 2 min with 35 °C restrictions. Samples were then pelleted by centrifugation and rinsed in 0.1 M cacodylate buffer (pH 7.2; Ted-Pella Inc.) 3X for 10 min each. Fixation for serial block-face scanning electron microscopy (SBFSEM) was conducted as described previously with slight modifications[66].

## SSC transplantation

To assess whether abnormalities in sperm morphogenesis are intrinsic to Arrdc5-deficient germ cells, germline transplantation was performed using the methodology described previously[30,60]. Briefly, testicular tissue was enzymatically digested with trypsin-EDTA solution (Sigma-Aldrich), Collagenase IV (Worthington) with DNase I (Sigma-Aldrich). Enzyme inactivation was achieved by adding 10% FBS (Gibco), and the cell suspensions passed through a 40 μm cell strainer. Next, cell suspensions were fractionated by centrifugation ($600 \times g$ for 8 min at 4 °C) through a 30% Percoll gradient, and the bottom fraction was collected to enrich for spermatogonial stem cells (SSCs), as described previously[60,67]. Cells were suspended in mouse serum-free media[67] at $8 \times 10^6$ cells/ml, and 5–10 μl was microinjected into the seminiferous tubules of *Nanos2*−/− recipient males that are ablated of endogenous germline[30]. For each recipient, one testis received cells from *Arrdc5*+/+ mice, and the contralateral testis received cells from *Arrdc5*−/− mice. At 3 months post-transplantation, testes and epididymis from recipients were excised following euthanasia and fixed in 4% PFA or Bouin's solution. Testes were then processed for paraffin embedding, and cross-sections were made for histological analysis. Epididymis from each recipient was flushed to collect sperm which were assessed for morphological defects.

## Quantification and statistical analysis

For scRNA-seq analysis, testicular tissue was collected from three different male pigs and three different male cattle. The downloaded mouse testis scRNA-seq data was generated from one animal. All quantitative comparisons between *Arrdc5*+/+ and *Arrdc5*−/− littermates were made from three to seven animals of each genotype. Quantitation of testis histology included at least 50 different seminiferous tubule cross-sections for three different males of each genotype.

Statistical analysis for scRNA-seq was conducted using R-Studio (v.3.4.4). For all other quantitative comparisons, statistical differences were determined using the two-tailed unpaired *t*-test and one-way or two-way ANOVA functions of GraphPad Prism software (v9.4.0 and v9.4.1).

## Reporting summary

Further information on research design is available in the Nature Portfolio Reporting Summary linked to this article.

## Data availability

Animal models generated by this study are available through requests of the corresponding author. The single-cell RNA-seq data generated in this study have been deposited in the Gene Expression Omnibus repository under accession ID GSE206156. Source data required for reanalysis are provided with this paper. Source data are provided with this paper.

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

## Acknowledgements

This work was supported by grant HD101223 awarded to J.M.O. from the Eunice Kennedy Shriver National Institute of Child Health and Human Development (NICHD). The authors thank personnel from the Franceschi Microscopy and Imaging Center at Washington State University for assistance with electron microscopy and imaging of sperm. We also thank the personnel of the Washington State University animal production and gene editing reagent cores for assistance with 10X-Genomics sample processing and generation of *Arrdc5-eGfp* mice. Lastly, we a grateful to all members of the Oatley laboratory for helpful discussions regarding the experimentation and manuscript.

## Author contributions

Conceptualization, M.I.G. and J.M.O.; Methodology, M.I.G., D.M., N.C.L., M.J.O., J.P., L.D.R., L.A.M., M.L.B., and J.M.O.; Validation, J.M.O.; Formal Analysis, M.I.G., N.C.L., J.P., L.D.R., and J.M.O.; Investigation, M.I.G., D.M., N.C.L., M.J.O., J.P., L.D.R., L.A.M., M.L.B., and J.M.O.; Resources, M.I.G., N.C.L., L.A.M., M.L.B., and J.M.O.; Writing—original draft, M.I.G. and J.M.O.; Visualization, M.I.G. and J.M.O.; Project administration, J.M.O.; Funding acquisition, J.M.O.

## Competing interests

The authors declare no competing interests.
