## [Peer Review File · Nature Communications]

ARRDC5 expression is conserved in mammalian testes and required for normal sperm morphogenesisREVIEWER COMMENTS

Reviewer #1 (Remarks to the Author):

The authors report that the α -arrestin ARRDC5 is required for spermiogenesis and male fertility in mice. They arrive at this conclusion using single cell RNAseq of testicular tissue from mice, pigs, and cattle to generate a multispecies integrated transcriptome database.

This dataset will be an important resource for research in male fertility.

Bioinformatic analysis was then used to generate a list of genes, that could be potential regulators of male fertility in these species. Of these, the authors choose the α -arrestin ARRDC5 for further analysis.

To begin the characterization of ARRDC5, they used CRISPR-Cas9 mediated gene editing to generate an ARRDC5 null allele. Loss of ARRDC5 did not disrupt mouse development and mice deficient for ARRDC5 did not show overt phenotypes.

Female ARRDC5^{-/-} were fertile whereas male ARRDC5^{-/-} were not. Loss of ARRDC5 disrupts critical steps in spermatogenesis. Moreover, strong evidence is presented that the function of ARRDC5 in spermatogenesis is intrinsic to germ cells. ARRDC5 was indeed only expressed in testicular tissue and is required in sperm development in particular for normal head morphogenesis, cytoskeletal reorganization, mitochondrial positioning, and flagellar development.

The findings clearly show that ARRDC5 is an essential regulator of sperm morphogenesis. While we are not experts for spermiogenesis and male fertility, the morphological and functional characterization of phenotype of the ARRDC5^{-/-} appears to be carefully executed, analyzed and interpreted.

The authors do not provide the mechanism of action for ARRDC5. Filling this gap will be a major task in the future.

There are only few minor comments:

1. Supplementary Figure 5E shows expression of Tnp1, whereas in the text it is stated as Tnp2 (Line 149 and 150)
2. Line 159: should read enhanced green fluorescent protein (eGFP)
3. Line 191: Body weight comparison between Arrdc5^{-/-} and wild-type littermates should be depicted in Supplementary Figure 10E. This figure does not exist.
4. Lines 379: 'While the E1 and E2 enzymes are universal in their functions, it is the family of E3 ligases that provide substrate specificity for targeted proteins in different cell types.' This statement is incorrect. In particular, for HECT ubiquitin ligases it is the adaptor proteins (such as α -arrestins) that determine substrate specificity.
5. We would like to point out that ARRDC5 is the only α -arrestin that lacks PPxY (or PY) motifs for the interaction with the WW3 domains of HECT type UB ligases. Hence at this point is unclear if ARRDC5 interacts with UB ligase at all. This should be considered in the discussion. It is therefore not likely that ARRDC5 function in spermiogenesis is independent of UB ligases.
6. In result-section please clarify that you have generated an ARRDC5-eGFP knock-in. Just reading the text left me wondering whether ARRDC5 was replaced with eGFP or if a ARRDC5-eGFP fusion protein

was generated. This is stated clearly in the respective supplementary figure legend, and should be made equally clear in the results section.

Reviewer #2 (Remarks to the Author):

The authors have undertaken a comprehensive study where they compare the testis transcriptomes across three species. This will provide useful information for use by the field. They have then selected one highly conserved gene, ARRDC5, to test the function of. ARRDC5 is required for male fertility and appears to play a role specifically in elongating sperm development. The phenotype resulting from the loss of ARRDC5 is reasonably well described (noting comments below). The data is broadly of a high quality and the manuscript well written. I do however note, and as detailed below, there are several places where the person/people who wrote the manuscript and prepared the figures have made small but significant errors. The figures do not always reflect what is written in the text.

Of greater significance, I wonder why germ cell ubiquitination was not studied to formally test the role of ARRDC5 in ubiquitination? Doing so would add some mechanistic data to the manuscript.

Specific comments

Title – I suspect that the term 'evolutionarily conserved' will bother career evolutionary biologists. As such I suggest that the title be changed to something like 'Arrdc5 is a conserved gene across mammals and is required for male fertility'.

Line 21 – OAT is technically not a diagnosis but rather a description of a clinical presentation.

Line 28 – add in the word mammalian before cell as there are other examples in other species.

Line 53 – the energy required for sperm tail function is provided by mitochondria (as stated) but also by the glycolysis pathway anchored to the fibrous sheath. The latter is absolutely critical in mouse sperm. Please update this sentence

In several places – use of the word stage should be checked, noting the strict meaning of 'stage' in spermatogenesis.

Fig. 4 – a tip. Staining sperm with haematoxylin and eosin is cheap and gives very nice results. I personally prefer it to Dip quick in that you can see more sub-structures.

Line 234 – delete the word syndrome as it is used incorrectly.

Line 237 – this section needs to be rewritten although I am not questioning the result. Capacitation and hyperactivation are biochemically separate processes. They usually occur in parallel, but they can be uncoupled. The ability to undergo the acrosome reaction is directly coupled to capacitation.

Line 251 – reword. PNA is not an acrosome component, rather it binds to components of the acrosome.

On a related topic, did you check that KO sperm have acrosomes in the first place ie. did non-ionophore challenged sperm stain positively with PNA? The IVF data could be consistent with the absence of an acrosome. Given that some of the IF data suggests ARRDC5 is localised to the acrosome region, this should be checked. Ditto while I can see some hooked nuclei in panel A -/- many sperm appear to have blunted nuclei suggesting an acrosome defect. Some high magnification sperm images (stained with H&E) might be helpful. The blunted heads can also be seen in 6B. Unfortunately, the TEM in 6C is not informative in relation to this question.

Fig. 5C – I found this an odd way to express the data. How was stage defined? How can you have reduced stage II but normal stage III. Personally, I would remove this panel and concentrate on describing/illustrating this pathology. Can I also suggest that someone with a lot of experience check the staging indicated in 5A – I disagree with some of the labels.

Fig. 6C – please indicate the defects listed in the text in the figures e.g. it is not possible to see misplaced axonemes, disorganised mitochondrial sheaths or the absence of the post-acrosomal region in the images chosen. These are key claims.

Line 326 – the images in Fig. 7C do not support that statement that there are multiple nuclei and a single axoneme. Remember that TEM shows a thin 2D section of a 3D object.

Fig. 8 – nice data.

Conclusion – I disagree with the statement that few testes specific genes when knockout have been shown to cause male sterility in isolation. There would be several hundred. Perhaps rewrite the sentence to more accurately reflect previous data. I appreciate this is a promising phenotype.

Reviewer #3 (Remarks to the Author):

In this manuscript, Giassetti et al., report an essential role of arrestin-domain containing 5 (Arrdc5) in mammalian spermatogenesis. The authors conducted single cell RNA-seq analyses of testicular cells from prepubertal mice, pig, and cattle and found that Arrdc5 is only expressed in germ cells in these species. Integrated analyses using ENCODE and human GTEx dataset suggest that this gene is testis specific. The authors then generated Arrdc5-eGfp and Arrdc5^{-/-} mouse lines to provide evidences that Arrdc5 is present in spermatids and plays an indispensable role in spermiogenesis. Development of spermatogonia, spermatocytes and haploid round spermatids are not disrupted, however, sperm morphology and function are severely affected by Arrdc5 deletion. The author concluded that Arrdc5 is a core regulator of spermatogenesis and a potential male contraceptive target. Although this study provides important data regarding the role of Arrdc5 in spermiogenesis, it contains significant flaws that fail to support major conclusion of this manuscript.

1. Although not published in peer reviewed journal, a publicly available report (International Mouse Phenotyping Consortium, MGI: 1924170) indicates that, in addition to male infertility, Arrdc5 knockout mice show abnormal retina morphology, increased hemoglobin, circulating fructosamine and other metabolites. The authors stated that loss of Arrdc5 causes male sterility without impacting other physiological processes, but did not provide any experimental evidence. For example, it is important to examine eye development because a male contraceptive target must not affect other important organs.
2. "testis specific" and "testis enriched" are completely different. Human GTEx data show that Arrdc5 is enriched in testis, but also present in whole blood and other tissues (Supplemental Fig.4). Unfortunately, whether these tissues (especially whole blood and retina) of mice, pig and cattle express Arrdc5 is not clear. Therefore, the conclusion that Arrdc5 in mammalian species from mice to human is germ cell specific lacks important evidence.
3. A big shortcoming is that this study only describes the infertile phenotype, although the authors state that Arrdc5 serves as a "core regulator". Without defined molecular mechanism, it is difficult to interpret the current data as "core" regulator of spermatogenesis. Similar defects have been reported in other knockout lines, is Arrdc5 functionally interacting with these proteins such as Capza3, or Ccdc62 to serve its core role?
4. SSC transplantation experiment was not necessary for this study. Only one sperm is shown in Fig.8B and it appears that elongating spermatids were entirely missing and SSC transplantation generated a completely different phenotype.

5. Arrdc5^{-/-} sperm fertilized nude oocytes and these zygotes developed to the blastocyst stage, these data indicate that Arrdc5^{-/-} sperm can produce normal embryos after ICSI. However, because severe DNA damage is observed in Arrdc5^{-/-} sperm, it will be interesting to examine whether these embryos can develop normally in vivo.
6. In human, has anyone detected Arrdc5 mutation in OAT patients?
7. Figure 4J shows the cleavage rate of control and Arrdc5 knockout sperm, but huge variation exists in control. Cleavage rate is from 20 to 85% for control. Why?
8. Figures 6 & 7 show TEM images for epididymal sperm of Arrdc5^{-/-} mice. Is microtubule cytoskeleton (9+2) normal?

Response to Referee Comments/Critiques

We are grateful to the reviewers for making encouraging comments and constructive criticisms. We have tried to address all critiques and suggestions for improvement by adding new data or revising the text and figures or explaining why we are unable to address a concern. All revisions to the manuscript text are highlighted in yellow. Overall, we feel that the scientific merit of the manuscript has been significantly improved.

Response to Reviewer 1

Summary Comments by the reviewer: The authors report that the α -arrestin ARRDC5 is required for spermiogenesis and male fertility in mice. They arrive at this conclusion using single cell RNAseq of testicular tissue from mice, pigs, and cattle to generate a multispecies integrated transcriptome database. This dataset will be an important resource for research in male fertility. Bioinformatic analysis was then used to generate a list of genes, that could be potential regulators of male fertility in these species. Of these, the authors choose the α -arrestin ARRDC5 for further analysis. To begin the characterization of ARRDC5, they used CRISPR-Cas9 mediated gene editing to generate an ARRDC5 null allele. Loss of ARRDC5 did not disrupt mouse development and mice deficient for ARRDC5 did not show overt phenotypes. Female ARRDC5^{-/-} were fertile whereas male ARRDC5^{-/-} were not. Loss of ARRDC5 disrupts critical steps in spermatogenesis. Moreover, strong evidence is presented that the function of ARRDC5 in spermatogenesis is intrinsic to germ cells. ARRDC5 was indeed only expressed in testicular tissue and is required in sperm development in particular for normal head morphogenesis, cytoskeletal reorganization, mitochondrial positioning, and flagellar development. The findings clearly show that ARRDC5 is an essential regulator of sperm morphogenesis. While we are not experts for spermiogenesis and male fertility, the morphological and functional characterization of phenotype of the ARRDC5^{-/-} appears to be carefully executed, analyzed and interpreted.

Reviewer Comment 1: The authors do not provide the mechanism of action for ARRDC5. Filling this gap will be a major task in the future.

Author Response: We agree with the reviewer's perspective and are aiming to elucidate mechanism of action with future research. Considering the uniqueness of ARRDC5 compared to other ARRDC family members, exploring mechanism of action is almost starting from scratch and as such new tools are required. This exploration will take significant effort, resources, and innovation which we feel is best addressed with future projects aimed specifically at defining functional features of ARRDC5 protein in male germ cells.

Reviewer Comment 2: Supplementary Figure 5E shows expression of Tnp1, whereas in the text it is stated as Tnp2 (Line 149 and 150)

Author Response: We thank the reviewer for catching this typo and it has been corrected.

Reviewer Comment 3: Line 159: should read enhanced green fluorescent protein (eGFP)

Author Response: We thank the reviewer for catching this typo and it has been corrected.

Reviewer Comment 4: Line 191: Body weight comparison between Arrdc5^{-/-} and wild-type littermates should be depicted in Supplementary Figure 10E. This figure does not exist.

Author Response: We thank the reviewer for catching that data was missing in the initial submission. We accidentally left out of the figure when creating the original submission. The data have now been included in the revised version of Figure 10E.

Reviewer Comment 5: Lines 379: 'While the E1 and E2 enzymes are universal in their functions, it is the family of E3 ligases that provide substrate specificity for targeted proteins in different cell types.' This statement is incorrect. In particular, for HECT ubiquitin ligases it is the adaptor proteins (such as α -arrestins) that determine substrate specificity.

Author Response: We thank the reviewer for pointing out the inaccuracy of this statement and have revised it as suggested.

Reviewer Comment 6: We would like to point out that ARRDC5 is the only α -arrestin that lacks PPxY (or PY) motifs for the interaction with the WW3 domains of HECT type UB ligases. Hence at this point is unclear if ARRDC5 interacts with UB ligase at all. This should be considered in the discussion. It is therefore likely that ARRDC5 function in spermiogenesis is independent of UB ligases.

Author Response: We appreciate the reviewer bringing up this point as it is important for postulating on the mechanism of action for ARRDC5 in regulating spermiogenesis. The discussion has been modified to point out that ARRDC5 is the only α -arrestin that lacks the PPxY motifs needed for interaction with HECT E3 ubiquitin ligases, and that the molecule is likely acting independent of this known mechanism of action for other α -arrestins.

Reviewer Comment 7: In result-section please clarify that you have generated an ARRDC5-eGFP knock-in. Just reading the text left me wondering whether ARRDC5 was replaced with eGFP or if a ARRDC5-eGFP fusion protein was generated. This is stated clearly in the respective supplementary figure legend and should be made equally clear in the results section.

Author Response: We thank the reviewer for pointing out the confusion and have revised the text to clarify that eGfp coding sequence was inserted into the *Arrdc5* locus, in particular that the knock-in replaced the stop codon of exon 3.

Response to Reviewer 2

Summary Comments by the reviewer: The authors have undertaken a comprehensive study where they compare the testis transcriptomes across three species. This will provide useful information for use by the field. They have then selected one highly conserved gene, ARRDC5, to test the function of. ARRDC5 is required for male fertility and appears to play a role specifically in elongating sperm development. The phenotype resulting from the loss of ARRDC5 is reasonably well described (noting comments below). The data is broadly of a high quality and the manuscript well written. I do however note, and as detailed below, there are several places where the person/people who wrote the manuscript and prepared the figures have made small but significant errors. The figures do not always reflect what is written in the text.

Reviewer Comment 1: Of greater significance, I wonder why germ cell ubiquitination was not studied to formally test the role of ARRDC5 in ubiquitination? Doing so would add some mechanistic data to the manuscript.

Author Response: We appreciate the reviewer's perspective on the value of adding some mechanistic data and agree that delving into mechanism of action for ARRDC5 in regulating spermatogenesis is an important next step. However, considering that ARRDC5 is unique in molecular structure compared to other α -arrestins, investigating mechanism of action is essentially starting from scratch and we feel that this task is best left to future experiments. Although other ARRDC molecules are known to be regulators of protein ubiquitination, as pointed out by reviewer 1, ARRDC5 lacks key features of the protein family such as PPxY motifs that are required for interaction with the abundant HECT type E3 ubiquitin ligases. Thus, if ARRDC5 has a role in ubiquitination, it will be unlike other α -arrestin molecules and likely require in-depth exploration to identify the types of E3 ubiquitin ligases that it interacts with and the specific proteins that are ubiquitinated by its actions. To properly investigate this aspect, we will need to define the protein interactome of ARRDC5 in spermatids. Going down this road is a large undertaking that we believe is out of scope for the current study and best left for future experiments.

Reviewer Comment 2: Title – I suspect that the term 'evolutionarily conserved' will bother career evolutionary biologists. As such I suggest that the title be changed to something like 'Arrdc5 is a conserved gene across mammals and is required for male fertility'.

Author Response: We thank the reviewer for making this suggestion and agree. In considering how to address it, we have revised the title to relate directly to the findings including application to mammals and sperm morphogenesis.

Reviewer Comment 3: Line 21 – OAT is technically not a diagnosis but rather a description of a clinical presentation.

Author Response: We thank the reviewer for pointing out this distinction and have revised the statement accordingly.

Reviewer Comment 4: Line 28 – add in the word mammalian before cell as there are other examples in other species.

Author Response: The revision has been made as suggested.

Reviewer Comment 5: Line 53 – the energy required for sperm tail function is provided by mitochondria (as stated) but also by the glycolysis pathway anchored to the fibrous sheath. The latter is absolutely critical in mouse sperm. Please update this sentence.

Author Response: The sentence has been updated as suggested.

Reviewer Comment 6: In several places – use of the word stage should be checked, noting the strict meaning of ‘stage’ in spermatogenesis.

Author Response: We thank the reviewer for pointing this out and have revised the use of stage for describing phases of germ cell development in spermatogenesis.

Reviewer Comment 7: Fig. 4 – a tip. Staining sperm with haematoxylin and eosin is cheap and gives very nice results. I personally prefer it to Dip quick in that you can see more sub-structures.

Author Response: We thank the reviewer for this advice and have used it to assess acrosomes on *Arrdc5*^{-/-} sperm in new data presented for supplementary Figure 11D.

Reviewer Comment 8: Line 234 – delete the word syndrome as it is used incorrectly.

Author Response: Edit has been made as suggested.

Reviewer Comment 9: Line 237 – this section needs to be rewritten although I am not questioning the result. Capacitation and hyperactivation are biochemically separate processes. They usually occur in parallel, but they can be uncoupled. The ability to undergo the acrosome reaction is directly coupled to capacitation.

Author Response: We appreciate the reviewer’s comments on how we described capacitation and hyperactivation and have attempted to revise the section to provide an accurate representation of the processes.

Reviewer Comment 10: Line 251 – reword. PNA is not an acrosome component, rather it binds to components of the acrosome.

Author Response: We thank the reviewer for pointing out this inaccuracy and have revised the text accordingly.

Reviewer Comment 11: On a related topic, did you check that KO sperm have acrosomes in the first place ie. did non-ionophore challenged sperm stain positively with PNA? The IVF data could be consistent with the absence of an acrosome. Given that some of the IF data suggests ARRDC5 is localised to the acrosome region, this should be checked. Ditto while I can see some hooked nuclei in panel A ^{-/-} many sperm appear to have blunted nuclei suggesting an acrosome defect. Some high magnification sperm images (stained with H&E) might be helpful. The blunted heads can also be seen in 6B. Unfortunately, the TEM in 6C is not informative in relation to this question.

Author Response: We appreciate the reviewer’s thoughts on whether the acrosome is intact for *Arrdc5* KO sperm. Considering the array of head morphology defects that occur with *Arrdc5* KO sperm, acrosome malformation is to be expected. However, malformation doesn’t

necessarily mean absence. Although clearly different compared to wild-type sperm, even with head malformations, we still observed PNA binding to *Arrdc5* KO sperm (Figure 4G). We found very few sperm from *Arrdc5* KO mice that did not have at least some PNA binding following ionophore incubation (quantified in Figure 4H). These data indicate that at least partial acrosome formation occurs with *Arrdc5* KO sperm. Because at least some PNA binding was observed for a majority of *Arrdc5* KO sperm following ionophore incubation, we have not performed staining without ionophore challenge. With the head malformities, we agree with the reviewer that it is unlikely that the acrosomes are functionally normal; this notion is supported by the fact that at least some PNA can bind to >90% following incubation with Ca²⁺ ionophore. We have revised the text to further clarify these points. In addition, we have taken the reviewer's advice and stained sperm with H&E to better visualize acrosomes. As can be seen in new supplementary Figure 11D, sperm from *Arrdc5*^{-/-} mice have elements of acrosomes but most are malformed which is to be expected given the array of abnormal head morphologies. Indeed, we observed sperm with enlarged heads that have malformed acrosomes, sperm with greatly misshapen heads that have remnants of an acrosome, and sperm with the midpiece wrapped around the head for which an acrosome is difficult to visualize. These new data have been discussed in the results section as well.

Reviewer Comment 12: Fig. 5C – I found this an odd way to express the data. How was stage defined? How can you have reduced stage II but normal stage III. Personally, I would remove this panel and concentrate on describing/illustrating this pathology. Can I also suggest that someone with a lot of experience check the staging indicated in 5A – I disagree with some of the labels.

Author Response: We thank the reviewer for bringing up these points and the suggestion of removing the staging data. However, we feel that including the data in Figure 5 is important because it provides indication that the initial steps in spermatogenesis are not defective in the absence of *Arrdc5*^{-/-} and the abnormality arises during late steps of spermiogenesis. We have taken the reviewer's thoughts into consideration, though, and have reassessed the staging in cross-sections of seminiferous tubules from *Arrdc5*^{+/+} and *Arrdc5*^{-/-} mice. Rather than breaking the data down by all 12 stages of the seminiferous cycle, we have grouped the stages as I-IV, V-VIII, and IX-XII. A new Fig. 5C has been added in the revised manuscript. In addition, we have relabeled the images in Fig. 5A based on the stage groupings. Moreover, the text of the results sections has been revised to account for the new data.

Reviewer Comment 13: Fig. 6C – please indicate the defects listed in the text in the figures e.g. it is not possible to see misplaced axonemes, disorganised mitochondrial sheaths or the absence of the post-acrosomal region in the images chosen. These are key claims.

Author Response: We have now labeled parts of images in Fig 6C to indicate examples of *Arrdc5*^{-/-} sperm with a disorganized mitochondrial sheath, absence of visible anterior acrosomes, and double midpieces. We agree with the reviewer that accurate visualization of misplaced axonemes is difficult in the images presented so have removed this from the text.

Reviewer Comment 14: Line 326 – the images in Fig. 7C do not support that statement that there are multiple nuclei and a single axoneme. Remember that TEM shows a thin 2D section of a 3D object.

Author Response: We agree with the reviewer that presence of multiple axonemes cannot be observed from the 2D section produced by TEM. Thus, the sentence has been revised to remove mention of this possible oddity.

Reviewer Comment 15: Fig. 8 – nice data.

Author Response: We thank the reviewer for this comment and agree that the data are important for demonstrating that impaired sperm morphogenesis of *Arrdc5*^{-/-} mice is intrinsic to the germ cells.

Reviewer Comment 16: Conclusion – I disagree with the statement that few testes specific genes when knockout have been shown to cause male sterility in isolation. There would be several hundred. Perhaps rewrite the sentence to more accurately reflect previous data. I appreciate this is a promising phenotype.

Author Response: We appreciate the reviewer's point but are unclear as to which statement in the discussion implies that few testes specific genes cause male sterility when knocked out. The one sentence that the reviewer may be referring to is, "*Few studies have uncovered genes expressed specifically in testicular germ cells that when inactivated lead to male specific sterility and even less have associated the expression of essential regulators identified in mice to evolutionary conservation in other mammalian species*". This statement is made to indicate that knowledge of genes expressed specifically in testicular germ cells with a critical role in spermatogenesis is limited. We certainly agree with the reviewer that a multitude of genes with testis-specific expression are known but believe that assignment to germ cells as intrinsic regulators of spermatogenesis is largely undefined. As suggested by the reviewer, we have attempted to rewrite the statement to more accurately reflect the state of understanding for the field of spermatogenesis.

Response to Reviewer 3

Summary Comments by the reviewer: In this manuscript, Giasseti et al., report an essential role of arrestin-domain containing 5 (*Arrdc5*) in mammalian spermatogenesis. The authors conducted single cell RNA-seq analyses of testicular cells from prepubertal mice, pig, and cattle and found that *Arrdc5* is only expressed in germ cells in these species. Integrated analyses using ENCODE and human GTEx dataset suggest that this gene is testis specific. The authors then generated *Arrdc5*-eGfp and *Arrdc5*^{-/-} mouse lines to provide evidences that *Arrdc5* is present in spermatids and plays an indispensable role in spermiogenesis. Development of spermatogonia, spermatocytes and haploid round spermatids are not disrupted, however, sperm morphology and function are severely affected by *Arrdc5* deletion. The author concluded that *Arrdc5* is a core regulator of spermatogenesis and a potential male contraceptive target. Although this study provides important data regarding the role of *Arrdc5* in spermiogenesis, it contains significant flaws that fail to support major conclusion of this manuscript.

Reviewer Comment 1: Although not published in peer reviewed journal, a publicly available report (International Mouse Phenotyping Consortium, MGI: 1924170) indicates that, in addition to male infertility, *Arrdc5* knockout mice show abnormal retina morphology, increased hemoglobin, circulating fructosamine and other metabolites. The authors stated that loss of *Arrdc5* causes male sterility without impacting other physiological processes, but did not provide any experimental evidence. For example, it is important to examine eye development because a male contraceptive target must not affect other important organs.

Author Response: We appreciate the reviewer's comment and became aware of the non peer-reviewed publicly available report of *Arrdc5* knockout mice after we had generated the CRISPR edited knockout described in the current manuscript. Although the standard IMPC phenotyping pipeline reported that the *Arrdc5* knockout mice generated by the Wellcome Trust Sanger Institute (WTSI) using ES cell targeting have an abnormal retina morphology, we have not observed overt defects in vision for any animals of the knockout line generated with CRISPR-Cas9 gene editing. Although our knockout animals have seemingly normal health, except for male specific sterility, which suggests that lack of ARRDC5 does not negatively impact other organs, we have not carried out in-depth assessment of all physiological systems. Thus, we have tempered the conclusionary statement to indicate that the findings demonstrate that ARRDC5 function is required specifically for male fertility and suggest that deficiency does not have overt negative impact on other physiological systems that impact health.

Regarding the reviewer's suggestion to examine eye development, we disagree that this assessment is needed for the current study for the following reasons:

1. New RT-PCR data has been included in the revised manuscript that demonstrates *Arrdc5* gene expression is absent in the eye of adult mice (Figure 2A in the revised manuscript, formerly Figure 2C). Although this finding does not assess whether *Arrdc5* is expressed at any point in fetal or perinatal development of the eye, it does demonstrate that expression is undetectable in the adult phase of life which is when a male contraceptive would be applied.
2. As best we can tell, mice from our knockout line do not have obvious visual impairment compared to wild-type littermates. Even out to over 1 year of age, the knockout animals retain seemingly normal eyesight and gross abnormalities of the eye are not observable.
3. We have included new body weight data for *Arrdc5* knockout mice at 3 and 16 weeks of age that shows no measurable difference compared to wild-type littermates (Supplementary

Figure 10E). These findings are a general indicator of normal physiology for *Arrdc5* knockout mice. If the vision and/or metabolism were negatively affected by lack of ARRDC5 expression, one would expect a difference in body weight by 4 months of age compared to wild-type counterparts.

Reviewer Comment 2: “testis specific” and “testis enriched” are completely different. Human GTEx data show that *Arrdc5* is enriched in testis, but also present in whole blood and other tissues (Supplemental Fig.4). Unfortunately, whether these tissues (especially whole blood and retina) of mice, pig and cattle express *Arrdc5* is not clear. Therefore, the conclusion that *Arrdc5* in mammalian species from mice to human is germ cell specific lacks important evidence.

Author Response: New RT-PCR data has been added to Figure 2A in the revised manuscript (formerly Figure 2C) that demonstrates *Arrdc5* gene expression is undetectable in both the eye and whole blood of adult mice. At this time, we do not have access to eye and whole blood of cattle or pigs so have not added to the gene expression analysis for these species. Regardless, we believe that the mouse tissue analysis is indicative of *Arrdc5* expression in other species and taken together with the cattle and pig tissue analyses support the conclusion of testis specificity.

Reviewer Comment 3: A big shortcoming is that this study only describes the infertile phenotype, although the authors state that *Arrdc5* serves as a “core regulator”. Without defined molecular mechanism, it is difficult to interpret the current data as “core” regulator of spermatogenesis. Similar defects have been reported in other knockout lines, is *Arrdc5* functionally interacting with these proteins such as *Capza3*, or *Ccdc62* to serve its core role?

Author Response: We agree with the reviewer that elucidating the mechanism of action for ARRDC5 in regulating spermatogenesis is a key next step. However, we feel that this exploration is best left for future studies that can focus on understanding the molecular function of ARRDC5. At present, the tools needed to define the protein interactome of ARRDC5 are not available. We have been unable to identify a commercially available primary antibody that is specific for ARRDC5 as shown in Supplemental Figure 6. Thus, to assess functional interaction between ARRDC5 and other molecules reported to have important roles in spermatogenesis such as *Capza3* and *Ccdc62*, new reagents or mouse models will need to be generated. Again, we believe that this will be an important next step for future studies. With that said, we do appreciate the reviewer’s concern about referring to ARRDC5 as a “core” regulator of spermatogenesis in the abstract. The phenotype of the knockout mouse model unequivocally shows that ARRDC5 plays an essential role in spermatogenesis which we consider to be core in its nature. However, we understand that the term core is considered differently amongst scientists and have revised the abstract to temper statements about the functional attributes of ARRDC5.

Reviewer Comment 4: SSC transplantation experiment was not necessary for this study. Only one sperm is shown in Fig.8B and it appears that elongating spermatids were entirely missing and SSC transplantation generated a completely different phenotype.

Author Response: Although we appreciate the reviewer’s viewpoint on this experiment and opinion of the outcomes, we disagree that it is not an important piece to the story. As we stated in the final section of the Results, the purpose of the SSC transplantation experiments was to “determine whether defects in sperm production of *Arrdc5*^{-/-} mice are intrinsic to germ cells”.

The SSC transplantation directly tests this, and we believe the outcomes demonstrate unequivocally that lack of ARRDC5 expression in germ cells is the cause of defective spermatogenesis. Even if the defective regenerated spermatogenesis originating from *Arrdc5*^{-/-} SSCs in a recipient testis appears different compared to steady-state spermatogenesis in an *Arrdc5*^{-/-} mouse testis, the conclusion that impaired capacity for normal spermatogenesis is intrinsic to ARRDC5-deficient germ cells is still supported. We have not made revision to the manuscript in response to this reviewer critique.

Reviewer Comment 5: *Arrdc5*^{-/-} sperm fertilized nude oocytes and these zygotes developed to the blastocyst stage, these data indicate that *Arrdc5*^{-/-} sperm can produce normal embryos after ICSI. However, because severe DNA damage is observed in *Arrdc5*^{-/-} sperm, it will be interesting to examine whether these embryos can develop normally *in vivo*.

Author Response: We agree with the reviewer that determining whether any embryos generated from *Arrdc5*^{-/-} sperm can develop normally *in vivo* is an important next step. However, we also want to point out that only 8% of embryos fertilized by *Arrdc5*^{-/-} sperm advanced to blastocyst stage *in vitro*. This is a very low percentage, thus the chances of obtaining a pregnancy by embryo transfer after ICSI is challenging. In new experiments, we have performed two rounds of ICSI with *Arrdc5*^{-/-} sperm and found that both the cleavage rate and blastocyst rate are reduced compared to wild-type sperm. Indeed, the *in vitro* blastocyst rate following ICSI with *Arrdc5*^{-/-} sperm is only 4%. These new data have been added to Supplemental Figure 12 of the revised manuscript. Based on the *in vitro* development rates, obtaining pregnancies from ICSI generated embryos would be very challenging. In our opinion, this step is outside the scope of the current study and best left to a future study that can focus on the outcomes of using sperm from *Arrdc5*^{-/-} males for assisted reproductive technologies.

Reviewer Comment 6: In human, has anyone detected *Arrdc5* mutation in OAT patients?

Author Response: We are not aware of evidence in peer-reviewed scientific literature that associates mutations in the *Arrdc5* gene to impaired spermatogenesis in men. However, to the best of our knowledge the level of GWAS studies on human males that are diagnosed with OAT is thin. Indeed, in comparison to the numerous studies that have been conducted on men with azoospermia (obstructive and/or nonobstructive), the data generated for men with OAT is quite limited. Thus, associating mutations in genes like *Arrdc5* to abnormal sperm morphogenesis in men is certainly an important element for the reproductive biology/medicine field to explore in future studies. We believe that our report of *Arrdc5* expression being enriched, if not specific, for testicular germ cells in multiple mammalian species and required for sperm morphogenesis in mice will provide impetus for investigators in the fields of spermatogenesis and male fertility to explore links with DNA mutations in men, domestic animals, and wildlife.

Reviewer Comment 7: Figure 4J shows the cleavage rate of control and *Arrdc5* knockout sperm, but huge variation exists in control. Cleavage rate is from 20 to 85% for control. Why?

Author Response: We appreciate the reviewer bringing up this point as the variation with wild-type sperm from this experiment is a bit outside of the norm. There are a multitude of possible reasons including biological variation or variations in standard reagents that were used for IVF. Regardless, the same conditions were applied to sperm from *Arrdc5*^{-/-} and *Arrdc5*^{+/+} mice. Although there was a wide variation in cleavage rate for *Arrdc5*^{+/+} sperm across the 6 different

IVF sessions that were performed for this study, 0 cleavage stage embryos were generated with sperm from any *Arrdc5*^{-/-} male. Thus, we stand by the conclusion drawn from these data that sperm of *Arrdc5*^{-/-} mice are unable to fertilize intact oocytes.

Reviewer Comment 8: Figures 6 & 7 show TEM images for epididymal sperm of *Arrdc5*^{-/-} mice. Is microtubule cytoskeleton (9+2) normal?

Author Response: This is an interesting question raised by the reviewer but unfortunately is difficult to fully address. As we have shown, ARRDC5 deficiency leads to an array of sperm abnormalities ranging from seemingly normal heads but with a bent neck to macrosperm to very misshapen heads to morphologically malformed midpieces and multiple axonemes. Based on these observations, it is logical to postulate that microtubule cytoskeleton structure will vary across the different sperm abnormalities. If there was a singular oddity to the sperm of *Arrdc5*^{-/-} mice, clearly assessing normalcy of the microtubule cytoskeleton would be straightforward. However, with the array of different abnormalities for sperm from *Arrdc5*^{-/-} mice, parsing out any abnormalities of microtubule cytoskeleton structure with different sperm abnormalities is almost a study unto itself. We believe that the reviewer's question is an important one and to properly address it requires deep characterization to fully assess microtubule structure in each different type of sperm abnormality observed for *Arrdc5*^{-/-} mice. In our opinion, this effort is out of scope for the current study and warrants addressing in future focused studies.

Reviewers' comments:

Reviewer #1 (Remarks to the Author):

The authors have addressed my initial concerns and the paper should be published - congratulations on a very interesting story

Reviewer #2 (Remarks to the Author):

I found this a disappointing revision in that very few of the revisions I suggested (beyond typographical) appear to have been seriously considered.

Specifically

Comment 1 - the request for mechanistic data was not addressed. This on its own may be understandable.

Comment 11 - acrosome formation and the request for pre-ionophore data. Re reviewers seem to be splitting hairs here. The presence of significant acrosome defects is important and should be considered more carefully. Noting, and contrary to what is in the text, acrosome formation initiates early in spermiogenesis suggesting ARRDC5 has a role early in haploid germ cell development. It is a pity the authors did not choose to explore this more through eg the inclusion of EM data as well as testing P4 and ionophore induced acrosome function.

Comment 12 - and related to above, the grouping of stages makes even less sense that attempting to group individual stage. I assume that the purpose to the analysis was to determine where defects first appear in KOs. This should be determined at a cell type (step) level and will likely require higher resolution than light level tissue histology. As per the point above, I suggest you look closely at acrosome formation. This will in form the IVF data.

Comment 13 - the decision to remove data and claims rather than replace the EM images is curious. If these abnormalities were significant, why not replace the images? Data to address the underlying cause of motility defects is now completely missing. The arrows indicating the anterior acrosome (bottom right) and the sperm nucleus are both pointing to head-tail-coupling (neck) region. Indeed what is meant by anterior acrosome is not clear at all. Do you mean the post-acrosomal region? Given these errors, it is difficult to have confidence in any of the EM data.

Comment 16 - the revised text is more confusing than the original version. Perhaps break it into two sentences. Do the authors mean

'Discovery of evolutionary conserved genes expressed specifically in testicular germ cells with an essential role in determining sperm function is uncommon.' If yes, as per my original comments I dispute this claim. There are many germ cell-enriched gene expression models that result in male infertility. The majority are expressed in multiple species, including beyond mammals.

These comments plus the realisation from other reviewer's comments that the mouse model did not result in tagged-ARRDC5 but rather GFP alone, has tempered my enthusiasm for this study.

As an aside, mouse vision cannot be accurately determined by watching behaviour in a standard mouse cage. <https://pubmed.ncbi.nlm.nih.gov/28760697/>

Response to Referee Comments/Critiques

We thank the reviewers for continued effort in trying to improve the merit and clarity of the manuscript. We have tried to address remaining concerns with revisions to the text and figures or explaining why a concern has not been addressed. All revisions to the manuscript text for the most recent round of review are highlighted in blue. Revisions made from the first round of review are highlighted in yellow. Overall, we believe strongly that the major findings presented in the manuscript are novel, advancing of knowledge, and fully supported by the data; these include, 1) expression of ARRDC5 is testis enriched, if not specific, in mice, pigs, cattle, and humans (this has never before been described), 2) ARRDC5 has an essential role in sperm morphogenesis (the knockout mouse model clearly demonstrates this and a role for ARRDC5 in mammals has not been described previously), and 3) sperm generated in the absence of ARRDC5 are unable to fertilize eggs and yield embryos (the mating trials, in vitro fertilization, and intracytoplasmic sperm injection data clearly support this and these findings have novel implications in assisted reproduction, male fertility assessment, and male contraceptive development). In addition, we believe the multispecies integrated testicular transcriptome database produced in this study will be of major utility to investigators in the fields of developmental and reproductive biology.

Response to Reviewer 2

Summary Comments by the reviewer: I found this a disappointing revision in that very few of the revisions I suggested (beyond typographical) appear to have been seriously considered.

Author Response: We appreciate the reviewer's perspective that the first revision fell short of expectation but assure the reviewer that we did take the suggestions seriously. In the first round of review, the reviewer made 16 different critiques/suggestions. We responded to all with inclusion of new data, revisions to figures, and revisions to the text. Of note, 3 of the 16 critiques/suggestions made by the reviewer asked for consideration of new data; the remaining 13 of 16 critiques/suggestions were addressable by revisions to the text only.

One of the critiques (previously Comment 1) asked for assessment of protein ubiquitination. As we explained before, this analysis is not straightforward because ARRDC5 lacks the key domain of other α -arrestins that are required for interaction with HECT type E3 ubiquitin ligases and therefore is unlikely to have a classical role in ubiquitination. This perspective was shared by Reviewer 1 who suggested that assessing the molecular function of ARRDC5 should be a goal of future studies.

Another request (previously Comment 11) for additional data to visualize acrosomes was "Some high magnification sperm images (stained with H&E) might be helpful". We have added these data to the manuscript as Supplemental Figure 11D.

The reviewer also asked if non-ionophore challenged *Arrdc5* KO sperm stain for PNA binding. Because ionophore treated sperm from the KO stain for PNA binding we struggled to understand how adding these data would fill a void in understanding. Thus, we did not include data on non-ionophore treated sperm in the first revision. However, we have now performed PNA binding and staining with non-ionophore treated sperm and observed no differences compared to post-ionophore treatments (new images provided as Supplemental Figure 12 and quantitation included in revised Figure 4G). We believe that these data support a conclusion that ionophore induced reaction of whatever acrosomal components are formed with *Arrdc5* KO sperm is significantly reduced compared to wild-type sperm.

The reviewer also made a statement that TEM images in Figure 6C are not informative (previously Comment 13). Because details of what was lacking in the images to render them uninformative were not provided, we struggled with how to address the comment but tried by adding labels and arrows to structures in the images that point out what we observed to be abnormal in *Arrdc5* KO sperm compared to wild-type sperm. In the second revision we have added more TEM images of the *Arrdc5* KO sperm as a new Supplemental Figure 14 that provides further evidence of abnormal sperm structures. Collectively, we believe that the images provide for Figure 6 and Supplementary Figure 14 provide sufficient evidence to support our conclusion that sperm generated in the absence of ARRDC5 function have an array of structural abnormalities.

Lastly, the reviewer made a comment (previously Comment 12) that data presented in Figure 5C was odd and disagreed with some of the seminiferous epithelium stage labels in images of Figure 5A. In response, we reassessed the stages in dozens of seminiferous tubule cross-sections and revised the data presentation to be what we consider more clearly interpretable by grouping stages rather than breaking them out individually. In the second revision we have again reevaluated the staging in dozens of cross-sections and the outcomes are similar to what was presented in other versions of the manuscript. In the second revised version, we applied statistical analyses to the dataset which revealed no difference between genotypes. This analysis has been included in the manuscript. Although not statistically different, we believe the data are still of value to readers for gaining an appreciation of spermatogenesis in testes of *Arrdc5* KO mice.

Reviewer Comment 1 - the request for mechanistic data was not addressed. This on its own may be understandable.

Author Response: In the first round of review, the reviewer suggested that we assess protein ubiquitination to provide some mechanistic data. As we responded in the initial review, assessing protein ubiquitination in *Arrdc5* KO germ cells is not straightforward and lacks strong reasoning. Again, delving into mechanism of action for ARRDC5 is difficult because the molecular structure is different compared to all other known α -arrestins, thus the exploration would require significant tool building and further experimentation. As such, we believe it is beyond the scope of the current study. Without knowing the protein interactome of ARRDC5, examining a role in ubiquitination is very difficult.

Reviewer Comment 11 - acrosome formation and the request for pre-ionophore data. Re reviewers seem to be splitting hairs here. The presence of significant acrosome defects is important and should be considered more carefully. Noting, and contrary to what is in the text, acrosome formation initiates early in spermiogenesis suggesting ARRDC5 has a role early in haploid germ cell development. It is a pity the authors did not chose to explore this more through eg the inclusion of EM data as well as testing P4 and ionophore induced acrsome function.

Author Response: We agree with the reviewer that deeper investigation on defective acrosome formation in *Arrdc5* KO mice is important but believe this is best left for future studies. The array of different sperm deformities that are observed for *Arrdc5* KO mice makes examination of a specific type challenging which we feel requires an independent study. Indeed, *Arrdc5* KO mice generate sperm with a normal appearing head morphology but with a neck and/or midpiece deformity, as well as sperm with enlarged heads and sperm with severely misshapen heads. Sperm with normal appearing head morphology possess seemingly intact acrosomal structure as

evidenced by imaging from PNA binding and staining (new Supplemental Figure 12). In addition, PNA binding can be observed for sperm with malformed heads, albeit with abnormal morphology. These observations suggest acrosome formation occurs at some level during morphogenesis of most, if not all, of the sperm in *Arrdc5* KO mice.

There are several possible causes of the abnormal sperm head morphology including defective chromatin compaction and improper shedding of cytoplasm. Both possibilities could cause abnormal acrosome biogenesis as collateral damage or acrosome biogenesis itself could be directly impaired. Parsing out these possibilities requires a focused independent study and, from our perspective, is beyond the scope of the current manuscript. However, we do appreciate the reviewer's interest in knowing more about acrosome biogenesis in *Arrdc5* KO sperm and have characterized this aspect further. We have now conducted pre-ionophore PNA binding assessment with *Arrdc5* KO sperm and characterized and quantified the observations (new data included in Supplementary Figure 12 and Figure 4H). In addition, we have repeated the post-ionophore treatment PNA binding assessment (new data in Figure 4H). Collectively, the outcomes of these assessments demonstrate that sperm heads from *Arrdc5*^{-/-} mice possess a range of acrosome morphologies from normal with no observable differences compared to wild-type sperm to abnormal which is to be expected considering the severe head malformation observed for a major portion of epididymal sperm. Quantitative comparison of total PNA labeled sperm heads between pre- and post-ionophore treatment revealed no difference for *Arrdc5* KO mice, unlike sperm from wild-type mice of which PNA binding was significantly reduced post-ionophore treatment. These new findings provide 1) better characterization of acrosome defects, and 2) further support our conclusion that the acrosome reaction capacity, even if the acrosomal vesicle is abnormally formed, for sperm generated in the absence of ARRDC5 function is impaired.

Reviewer Comment 12 - and related to above, the grouping of stages makes even less sense that attempting to group individual stage. I assume that the purpose to the analysis was to determine where defects first appear in KOs. This should be determined at a cell type (step) level and will likely require higher resolution than light level tissue histology. As per the point above, I suggest you look closely at acrosome formation. This will in form the IVF data.

Author Response: Our goal for conducting the staging analysis was to provide another piece of information for whether spermatogenesis overall is disrupted in the absence of ARRDC5 function. The analysis was not intended to determine where defects first appear or relate it to acrosome formation. Indeed, we draw no conclusions about acrosome biogenesis from the data. We agree with the reviewer that close examination at the cell type (i.e. spermatid step) level is needed to clearly understand where defects in spermiogenesis initiate but as discussed previously, feel that this depth of analysis is out of scope for the current study and properly addressing it is best left to future studies due to the complexity of sperm abnormalities that occur in *Arrdc5* KO mice. Indeed, an array of different sperm deformities are observed for *Arrdc5* KO mice ranging from morphologically normal heads but with bent necks to severely malformed heads. Linking which spermatids in the seminiferous epithelium to sperm that will have specific structural defects would require significant further investigation. For this reason, we do not believe examination of spermatids histologically for defining when impairments in acrosome biogenesis may initiate is plausible. To do this will require associating specific characteristics or lack thereof with propensity to yield sperm with specific deformities and this will require a focused independent study.

To address validity of the staging analysis, we have again reanalyzed cross-sections of testes from *Arrdc5*^{-/-} and *Arrdc5*^{+/+} littermates for distribution of stages of the seminiferous cycle.

Regardless of whether the data are presented as all 12 stages or as groupings of stages there is still misalignment between the two genotypes. The data were generated in a blinded and unbiased manner; thus, we believe the findings are real and conclusions are supported. In the second revised version of the manuscript, we have applied statistical analysis to the dataset and found the differences between genotypes is not significant. This information has been incorporated into the manuscript. Although not significantly different, we still believe the data are important for providing readers with an overall assessment of spermatogenesis in testes of *Arrdc5* KO mice.

Reviewer Comment 13 - the decision to remove data and claims rather than replace the EM images is curious. If these abnormalities were significant, why not replace the images? Data to address the underlying cause of motility defects is now completely missing. The arrows indicating the anterior acrosome (bottom right) and the sperm nucleus are both pointing to head-tail-coupling (neck) region. Indeed what is meant by anterior acrosome is not clear at all. Do you mean the post-acrosomal region? Given these errors, it is difficult to have confidence in any of the EM data.

Author Response: We appreciate the reviewer's perspective but are unsure what is being referred to as removed data. Perhaps the reviewer is referring to the revised way we are reporting data on distribution of stages of the seminiferous epithelium. If so, all the original data is still represented but grouped into broader stages. We revised the data reporting in response to the reviewer's comment that the original data presentation as all 12 stages of the seminiferous cycle was difficult to understand. We believe the staging data to be accurate and appreciate that different interpretations can be made based on differences in perspectives. However, we also feel that our interpretations of the data that a specific stage could not be applied to ~8% of seminiferous tubules cross-sections from testes of *Arrdc5* KO mice and because of this distribution of cross-sections as stage groupings was misaligned compared to wild-type control mice are supported by the data.

Regarding the removal of claims, perhaps the reviewer is referring to the sentence we removed from the text in response to suggestion from the reviewer; however, that sentence was speculation and not an interpretation or conclusion drawn from the data.

Regarding why we did not replace EM images for Figure 6C, in the first round of review the reviewer asked to "please indicate the defects listed in the text in the figures" which we did in the revision by adding indications of observed abnormalities to the images. In the second revision, we have included new TEM images as Supplementary Figure 14 that we feel provide further evidence of the array of structural defects for *Arrdc5* KO sperm. Overall, we believe the EM images clearly show abnormalities for sperm from *Arrdc5* KO mice compared to wild-type controls and our observations of disorganized mitochondrial sheath, abnormal post-acrosomal segment, and malformed midpiece are supported.

Regarding the now complete absence of data to address the underlying cause of motility defects, we are unsure what the reviewer is referring to. In neither the initial submission nor the revised manuscript did we present data to explain the underlying cause of motility defects. To do so would require significant further experimentation as there could be an array of possibilities. We believe the data presented clearly demonstrate that sperm from *Arrdc5* KO mice have a motility defect but delving into the underlying cause(s) is best left to future studies and is beyond the scope of the current manuscript.

Regarding errors in labels applied to the EM images that make it difficult to have confidence in the data, again we are unclear what the reviewer is referring to. For Figure 6C, we have not

included arrows indicating sperm nuclei and the arrow pointing to the acrosome was intended to show a misshapen sperm head where there is clearly not a post-acrosomal segment. Our intent with using the term “anterior acrosome” was to describe the outer acrosomal membrane on the anterior portion of the sperm head. Perhaps this is not the appropriate description; however other studies have referred to the acrosome covering the anterior portion of the sperm head as the anterior acrosome. In the results section, we had described the observation as sperm lacking a post-acrosomal segment, not an anterior acrosome. For consistency and accuracy, we have relabeled the images in Figure 6C with post-acrosomal segment or PS in place of anterior acrosome or AA. Overall, we believe that the EM images accurately reflect abnormalities between wild-type and *Arrdc5* KO sperm.

Reviewer Comment 16 - the revised text is more confusing than the original version. Perhaps break it into two sentence. Do the authors mean 'Discovery of evolutionary conserved genes expressed specifically in testicular germ cells with an essential role in determining sperm function is uncommon.' If yes, as per my original comments I dispute this claim. There are many germ cell-enriched gene expression models that result in male infertility. The majority are expressed in multiple species, including beyond mammals.

Author Response: We appreciate the reviewer’s opinion on this statement and believe the concern being voiced reflects differences in professional perspectives. Our statement was, “Discovery of genes expressed specifically in testicular germ cells that have an essential intrinsic role in regulating spermatogenesis and functional translation from discovery in mice to evolutionary conservation in other mammalian species has been limited”. We have not stated, nor do we intend to imply, that it is uncommon to discover genes expressed specifically in testicular germ cells with a role in sperm function. We stand by the perspective that discovery of genes with testicular germ cell specific expression that have intrinsic roles in sperm formation for mammalian species beyond the mouse has been limited. We note that germ cell-enriched and germ cell-specific do not have identical meanings. We don’t disagree that there are models of genes that are germ cell-enriched that have conserved roles in sperm function. However, the infertility in many of the models is not necessarily do specifically to an intrinsic role in germ cells as expression of the gene also occurs in some male reproductive tract somatic cells. Also, examples of demonstrated functional roles in mammals beyond the mouse are indeed limited. To address the reviewer’s concern, we have attempted to revise the statement for clarity.

New Reviewer Comment: These comments plus the realisation from other reviewer's comments that the mouse model did not result in tagged-AARDC5 but rather GFP alone, has tempered my enthusiasm for this study.

Author Response: The *Arrdc5-eGfp* model was generated to assess the testicular cell types that ARRDC5 protein is present in. This model produces a fused *Arrdc5-eGfp* transcript and the EGFP portion is liberated during translation. Thus, imaging for EGFP provides an accurate representation of the cell types that possess ARRDC5 protein. This approach has been used by a multitude of previous studies to accurately assess the cell types within a tissue that the protein product of a gene of interest is present in. Overall, we believe the model produced in this study is of major value and the imaging conducted with it provides important insights into the cell type expression profile for ARRDC5.

New Reviewer Comment: As an aside, mouse vision cannot be accurately determined by watching behaviour in a standard mouse cage. <https://pubmed.ncbi.nlm.nih.gov/28760697/>

Author Response: We appreciate this comment but have not made claims in the manuscript that watching behavior is an accurate means to assess vision in mice. We have clearly shown that *Arrdc5* is not expressed in the eye of adult mice, thus have no reason to suspect that the KO animals have impaired vision.